# Microphysical view of development and ice production of mid-latitude stratiform clouds with embedded convection during an extratropical cyclone

Yuanmou Du[1,3], Dantong Liu[2], Delong Zhao[1,3], Mengyu Huang[1,3], Ping Tian[1,3], Dian Wen[1,3], Wei Xiao[1,3], Wei Zhou[1,3], Hui He[1,3], Baiwan Pan[2], Dongfei Zuo[4], Xiange Liu[1,3], Yingying Jing[1,3], Rong Zhang[4], Jiujiang Sheng[1,3], Fei Wang[1,3], Yu Huang[1,3], Yunbo Chen[1,3], Deping Ding[1,3]

[1]Beijing Weather Modification Center, Beijing, 100089, China
[2]Department of Atmospheric Sciences, School of Earth Sciences, Zhejiang University, Hangzhou, Zhejiang, 310027, China
[3]Beijing Key Laboratory of Cloud, Precipitation and Atmospheric Water Resources, Beijing, 100089, China
[4]CMA Weather Modification Center, Beijing, 100081, China

*Correspondence to*: dantongliu@zju.edu.cn

**Abstract.** The microphysical properties associated with ice production importantly determine the precipitation rate of clouds. In this study, the microphysical properties, including the size distribution and particle morphology of water and ice, of stratiform clouds with embedded convection during an extratropical cyclone over northern China were characterized in situ. Stages of clouds, including young cells rich in liquid water and developing and mature stages with high number concentrations of ice particles ($N_{Ice}$), were investigated. The $N_{Ice}$ could reach 300 L$^{-1}$ in the mature stage, approximately two orders of magnitude higher than the primary ice number concentration calculated from ice nucleation. This high $N_{Ice}$ occurred at approximately −5 to −12°C, spanning the temperature region of the Hallett–Mossop process and possibly other mechanisms for secondary ice production (SIP). The $N_{Ice}$ was positively associated with the number concentrations of graupel with diameters ($d$) > 250 μm and large supercooled droplets ($d$ > 50 μm). The SIP rate was 0.005–1.8 L$^{-1}$s$^{-1}$, which was derived from the measured $N_{Ice}$ and the known ice growth rate between two sizes. The SIP rate could be produced by a simplified collision-coalescence model within an uncertainty factor of 5, by considering the collection of large droplets by graupel. The collection efficiency between the graupel and droplet was found to increase when the size of the droplet approached that of the graupel, which may improve the agreement between the measurements and the model. Importantly, the overall $N_{Ice}$ was found to be highly related to the distance to cloud-top (DCT). The level with a larger DCT had more rimed graupel particles falling from upper levels, which promoted coalescence processes between graupel particles and droplets, producing a greater fraction of smaller ice through SIP. This seeder-feeder process extended the avalanche SIP process at lower temperatures to −14°C, beyond the temperature region of the Hallett–Mossop process. The results illustrate the microphysical properties of clouds with convective cells at different stages, which will improve the understanding of the key processes in controlling the cloud glaciation and precipitation processes.

# 1 Introduction

Mid-latitude clouds generally contain a mixture of phases (Mülmenstädt et al., 2015). The microphysical properties associated with ice production or the conversion from liquid water to ice strongly determine the precipitation rate and lifetime of such clouds (Lau and Wu, 2003; Cantrell and Heymsfield, 2005). The growth rate of hydrometeors through the ice phase is usually greater than that through the warm rain phase (Lohmann and Feichter, 2005; Mcfarquhar et al., 2017). Understanding the ice production and glaciation processes in clouds is important for the accurate parameterization of microphysical processes in weather prediction models (Korolev et al., 2017; Bacer et al., 2021), and these processes need to be understood in the vertical dimension and during different stages of cloud development (Zhao et al., 2019).

In addition to the primary ice produced by homogeneous and heterogeneous nucleation processes from aerosol particles (Kanji et al., 2017), the secondary ice production (SIP) process can rapidly increase the amount of ice, reaching totals several orders of magnitude greater than the amount produced via ice nucleation (Mossop, 1985; Harris-Hobbs and Cooper, 1987; Field et al., 2016; Korolev et al., 2022). Consequently, SIP is an important process that accelerates cloud glaciation. SIP can occur under different ambient temperatures through different processes, including the rime splintering process, fragmentation during droplet freezing, fragmentation due to ice-ice collision, ice particle fragmentation due to thermal shock, fragmentation of sublimating ice particles, and the activation of ice-nucleating particles in transient supersaturation around freezing drops (Korolev et al., 2020). The Hallett–Mossop (H–M) mechanism has been well reproduced through laboratory work (Hallett and Mossop, 1974) and is usually introduced to explain the high ice number concentration at slightly subfreezing temperatures (−3 to −8 °C) (Hogan et al., 2002; Huang et al., 2008; Crosier et al., 2013; Korolev et al., 2022). The freshly formed ice generated via the H–M process exists mostly in the form of columns or needles (Locatelli et al., 2008; Crosier et al., 2011; Lloyd et al., 2014; Taylor et al., 2016), which is consistent with the diffusion growth habit of ice at such temperatures. However, the H–M process does not sufficiently explain the rapid SIP rate in observations, and fragmentation during droplet freezing and ice–ice collision may result in the production of high ice concentrations (Rangno and Hobbs, 2001). Furthermore, columns or needles may also be formed when ice from outside the H-M temperature zone is transported into the zone and subsequently grows (Field et al., 2016). Supercooled large drops may play important roles in the SIP process, as they can fracture when freezing and emit ice splinters (Lawson et al., 2015); this process could extend SIP to lower temperatures under the influence of strong updrafts. A recent study also revealed that the SIP process can occur at temperatures as low as −27°C (Korolev et al., 2022).

Mid-latitude clouds associated with extratropical cyclones are the main sources of precipitation in East Asia (Li et al., 2016). The microphysical properties of clouds over the North China Plain have been observed during frontal systems (Yang et al., 2017; Hou et al., 2021; Hou et al., 2023). More ice particles were found close to the convective region, and SIP was found to produce ice number concentrations of more than 300 L$^{-1}$, which may increase the intensity of precipitation. These studies suggest that the SIP process may be explained by the H–M process or by other mechanisms, such as collisional fragmentation, which may contribute to SIP in regions that do not fit the H–M criteria (Hou et al., 2023). However, the key

factors in controlling the SIP process and how these factors can influence SIP at different cloud stages have not been elucidated.

The cold front system formed by the merging of cold air from the rear of extratropical cyclones with the warm air mass brought in by the southwest warm and moist air along the edge of the subtropical high-pressure system is the main type of weather system that produces rainfall in northern China (Wang et al., 2014). This study investigates the microphysical properties of mid-latitude clouds formed via this typical weather system over the North China Plain through aircraft-based in situ measurements. Stages of clouds, including young cells rich in liquid water and developing and mature stages with high number concentrations of ice particles, were investigated. The key factors in controlling SIP are elucidated through calculations from measurements and modelling.

## 2 Experiment

### 2.1 Instrumentation

The King Air 350 aircraft of the Beijing Weather Modification Center was employed for in situ measurements in this work (Liu et al., 2020; Tian et al., 2020; Zuo et al., 2023). This study aimed to conduct continuous aircraft observations of clouds produced by an extratropical cyclone over northern China. The goals were to obtain in situ microphysical data on clouds during the development of a frontal system and to study the production of ice in clouds. The experiment was designed on the basis of numerical model forecasting results and real-time radar data. To capture the microphysical characteristics of stratiform clouds with embedded convection at various development stages, aircraft observations were made in accordance with real-time changes in precipitation radar echoes.

The air temperature was measured by a Rosemount total-air temperature probe (Lenschow and Pennell, 1974; Lawson and Cooper, 1990). The temperature may be underestimated because of water evaporation; however, this artefact is negligible for supercooled clouds (Lawson and Rodi, 1992; Korolev and Isaac, 2006), and no temperature shift in and out of clouds was observed in this study. The wind speed and wind direction were measured by an Aircraft Integrated Meteorological Measurement System (AIMMS20, Aventech Research, Inc.) with a temporal resolution of 1 second (Beswick et al., 2008). The distribution of aerosol particles ranging from 0.1 to 3 μm in diameter was measured by a passive cavity aerosol spectrometer probe (PCASP, DMT, Inc.) with a temporal resolution of 1 second (Cai et al., 2013).

A fast cloud droplet probe (FCDP, SPEC, Inc.) (Lance et al., 2010) was used to measure cloud droplets with a diameter range of 2–50 μm and had a resolution of approximately 3 μm. The FCDP resolves the particles into 20 size bins, and the optical sizing was calibrated with standard glass beads of known size. The liquid water content (LWC) for droplets with diameters of 2–50 μm was calculated by integrating the volume across all size bins from the FCDP (Lu et al., 2012). A two-dimensional stereoscopic optical array imaging probe (2D-S, SPEC, Inc.) was used to record images of cloud particles and determine their size, shape and concentration. 2D-S has two orthogonal laser beams that cross in the middle of the sample volume and cast shadowgraphs of the particles on two linear 128-photodiode arrays as particles transit through the laser

beams (Lawson et al., 2006). It can measure particles 10–1280 μm in diameter with a resolution of 10 μm and provides detailed information on liquid and ice-phase particles. The precipitation particles were measured by a high-volume precipitation spectrometer (HVPS, SPEC, Inc.) (Lawson et al., 1998), which is also an imaging array probe with a measurement range and resolution of 150–19200 μm and 150 μm, respectively. The laser beam of the HVPS illuminates the imaging system and records shadow images on a 128-element linear photodiode array as particles passes through the sample volume. An S-band weather radar located in Beijing (Jiang and Liu, 2014), which can detect targets within a radius of 230 km with a temporal and radial spatial resolution of 6 minutes and 1 km, respectively, was used to help analyse the macroscopic characteristics of the clouds. The distance from the radar to the observed cloud system in this study was approximately 50–200 km.

Optical array shadow imaging software was used to process the raw data from the 2D-S and HVPS. This software can distinguish between liquid drops and ice particles according to the circularity of the particles ($C$) (Crosier et al., 2011). $C$ is calculated with Eq. (1):

$$C = \frac{P^2}{4\pi A},\tag{1}$$

where P and A are the perimeter around the edge of the particle and the total area of the particle, respectively. A perfect sphere has a circularity of 1, and the other shapes have greater circularity. Irregular particles with greater circularity are considered ice particles because the shape of an ice particle is unlikely to be round. Considering that poorly imaged or distorted large drops/drizzle particles may be counted as ice particles, the circularity threshold for ice particles (irregular class) is set to 1.2. The calculated circularity values may also be less than 1 because the images are composed of only a small number of pixels, so the lower threshold for water drops (round class) is set to 0.9. In practical terms, particles with areas less than 20 pixels are classified into the small class because it is difficult to determine the shape of a particle spanning only a few pixels, and particles with areas greater than 20 pixels are classified into the round class ($0.9 \leq C < 1.2$) or irregular class ($C \geq 1.2$). The round and irregular classes are regarded as liquid drops and ice particles, respectively. The round class is composed of large droplets with diameters greater than 50 μm, which are referred to as large droplets in this study to distinguish them from the droplets (2–50 μm) measured by the FCDP.

The shapes of the irregular ice particles were further categorized into five habit classes, namely, linear, plate, irregular, aggregate and dendrite, according to the maximum dimension, width, linearity, circularity and density of the particles (Zhang et al., 2021). The mass of ice was determined by particle shape according to the approximate mass formulas for ice particles (Holroyd, 1987), and the ice water content (IWC) was subsequently calculated. It should be noted that the error in calculating the ice mass according to the mass–dimension relationship will increase when the ice particle size is larger and the shape has large irregularity to be classified (Crosier et al., 2013). The total water content (TWC) was obtained by adding the IWC calculated from the 2D-S (diameter 10–1280 μm) and the LWC measured by the FCDP (diameter 2–50 μm).

The concentration of ice nucleating particles (INPs) in this study was calculated via the following parameterization relationship (Demott et al., 2010):

$$n_{\text{IN},T_k} = a(273.16 - T_k)^b (n_{\text{aer},0.5})^{(c(273.16 - T_k) + e)}, \tag{2}$$

where $a = 0.0000594$, $b = 3.33$, $c = 0.0264$, and $e = 0.0033$. In this equation, $n_{\text{IN},T_k}$ represents the number concentration of INPs (L$^{-1}$), $T_k$ represents the temperature of the cloud in degrees Kelvin, and $n_{\text{aer},0.5}$ represents the number concentration of aerosol particles with diameters larger than 0.5 μm. In this study, the PCASP measurement was conducted below the cloud base, and the in-cloud PCASP data were excluded from the analysis because of cloud particles shattering on the inlet. Therefore, $n_{\text{aer},0.5}$ measured by PCASP below the cloud base was used for calculation.

## 2.2 Overview of the experiment

On September 26[th], 2017, light precipitation occurred in North China under the influence of an eastwards-moving upper trough. ERA5 reanalysis data at 08:00 BJT (UTC+8 h) with a resolution of 0.25 degrees from the European Centre for Medium-Range Weather Forecasts (ECMWF) revealed a deep cold vortex system in northwestern East Asia at 500 hPa (Fig. 1a), the bottom of which split into a shortwave trough and moved eastwards, leading to southwards-moving cold advection at the middle level and conditional instability stratification (Fig. S1). Figure 1b shows the existence of a convergence zone at 850 hPa, where a cold front was located, and sufficient water vapour was transported through the prefrontal southerly wind. The abundance of water vapour and upwards air motion led to the generation of a series of stratiform clouds, and convective clouds appeared under the condition of instability stratification.

The study region in this research was Zhangjiakou, Hebei Province (northwest of Beijing) and Beijing, and aircraft departed from the airport in northern Beijing at 09:54 BJT and flew to Zhangjiakou. The precipitation mainly occurred in Zhangjiakou and became weaker after 11:00 BJT on September 26[th], 2017; then, the precipitation band gradually moved to Beijing, and weather station observation data indicated that the precipitation rates during this experiment were generally less than 1 mm/h. Figure S2 shows the movement of the surface cold front, i.e., the convergence zone of cold and warm air masses at the surface as determined by the temperature and wind shear measured by the ground sites. The centre of the extratropical cyclone was located in Outer Manchuria (Fig. S3), and the surface cold front extended southwestwards from the position of the extratropical cyclone to the experimental region. The experimental region was situated within the trailing end of the cold front cloud system, which extended southwards from the extratropical cyclone cloud system. From 09:00 to 12:00 BJT, the surface cold front continued to move southeastwards and lifted the warmer and moist air mass in front. The warm air mass ascended along the front, forming clouds and precipitation, and the aircraft observation area was situated behind the cold front. The aircraft sampled the clouds formed in this cyclonic system at this stage, i.e., behind the surface cold front line (Fig. S3), including the newly formed, developing and mature clouds. This is a typical cloud system formed in such extratropical weather systems over northern China.

## 3 Results

### 3.1 Identifying stages of cloud development

Four relative stages during the lifecycle of clouds were identified during the experiment: developing (P1), mature (P2), dissipating (P3) and young cells (P4) in the cloud system, according to the different extents of cloud glaciation. The ice mass fraction ($F_{Ice}$: IWC/TWC) was used to indicate the different cloud development stages (Fig. 2) by considering that a more mature cloud has a greater glaciated fraction, for the discussions of cloud microphysics at different stages. Although the extents of glaciation between P2 and P3 were similar, P3 presented a narrower cloud band (Fig. 3) and a lower cloud-top (Fig. 4) for dissipating cells compared to the mature clouds in P2. The cloud system was formed through the combined effects of dynamic forcings induced by frontal uplift and moisture transport provided by the prefrontal southerly air mass. Therefore, this study postulated that the continuous clouds within the cloud system had similar dynamic and thermodynamic properties. Previous studies also pointed out the exchangeability between temporospatial domains of cloud properties in the same cloud system, where the properties and evolution of individual clouds were similar (Lensky and Rosenfeld, 2006; Yuan et al., 2010; Coopman et al., 2020). The aircraft was flown at flight altitudes of 3.2–5.7 km in P1, 5.2–5.8 km in P2, 4.9–5.2 km in P3 and 2.1–4.9 km in P4, and temperature and AIMMS data indicated that the 0°C layer was at approximately 3.4 km. The flight tracks mapping on the composite reflectivity of the precipitation radar data are shown in Fig. 3, coloured by the LWC from the FCDP and the IWC from the 2D-S. The radar times and the flight time windows for the four stages are shown in Table S1. In developing cells, a substantial LWC was detected, with values up to 0.3 g m$^{-3}$, and the aircraft penetrated a high IWC region in this cloud at 10:09-10:11 BJT, with the highest IWC exceeding 2 g m$^{-3}$ (Fig. 3a1, b1). Additionally, the $F_{Ice}$ in this stage ranged from zero (pure water) to one (pure ice) (Fig. 2). In the mature cells, $F_{Ice}$ ranged from 0.36–1, and the IWC generally exceeded 0.3 g m$^{-3}$ (Figs. 2 and 3b2). The maximum radar reflectivity of the mature cells increased from 20 dBZ to 27 dBZ at 10:06 to 10:42 BJT (Fig. 3a1, a2). During the dissipating stage, the ice-phase precipitation process occurred, and the radar reflectivity became weaker with a narrowed cloud band (Fig. 3a3, b3). The range of $F_{Ice}$ reached 0.56-1 (Fig. 2). The last stage was young cells with a lower glaciated fraction (Fig. 2), and the abundant liquid water produced from the newly developed thermals after the front cloud bands dissipated (Fig. 3a4, b4).

Figure 4 shows the microphysical properties of clouds and meteorological parameters in the four stages along the flight track. The vertical wind data during aircraft turns were excluded from Fig. 4 and were not used for the analysis. The cross section of radar reflectivity in Fig. 4a can provide information about the relative positions of the aircraft with respect to the cloud top and base, as well as the echo intensity of the cloud. The cross section of radar reflectivity along the flight track was calculated on the basis of the aircraft position. A vertical line was first determined according to the latitude and longitude of the aircraft; then, the azimuth angles, elevation angles, and range bins of equidistant points with a resolution of 30 m in the vertical direction were obtained. The radar reflectivity of each equidistant point was calculated using the nearest-neighbor scheme combined with a linear interpolation in vertical direction (NVI). A radar profile with a vertical resolution of 30 m along the flight track was obtained. The cloud-top height is indicated by the red line on the radar profile, and the areas with

radar reflectivity factors greater than or equal to 5 dBZ are considered clouds and other areas are considered clutter. This might provide a lower estimate of the cloud-top height because the rain radar was sensitive only to clouds with precipitation

and might not efficiently detected clouds dominated by liquid water. The size spectrum of the ice showed a bimodal mode with a minimum diameter ($d$) of 180 μm (Fig. S4). The fraction of smaller ice particles with $d < 180$ μm ($F_{smaller\ ice}$) was defined to imply the freshly formed smaller ice which had not experienced sufficient growth (Fig. 4b). The sensitivity was tested by altering the threshold from 160 to 200 μm, and the resulting difference in the smaller ice fraction was within 10%.

P1 featured strong updrafts with vertical wind speeds of up to 8.9 m/s, and the strong updraft region was dominated by ice

and precipitation particles (Fig. 4c-e). The low LWC in the strong vertical updraft may be caused by the rapid production of ice particles, which has also been observed in highly convective regions in the tropics (Lawson et al., 2015). The ice number peaked at a valley between two peaks of liquid water, but it is difficult to determine the vertical wind at the peak ice number due to aircraft turns (Fig. 4). However, in the subsequent level flight, a high ice number concentration ($> 170$ L$^{-1}$) was also observed in the strong updraft region. After the high ice number region, an LWC of up to 0.28 g m$^{-3}$ was observed in the

region with weaker updrafts.

The cloud-top height in P2 reached 10 km (Fig. 4a), which was the highest cloud-top among the clouds observed during the experiment. The LWC in P2 was considerably lower than that in P1, while there were more large droplets and ice particles in the clouds (Fig. 4d, e). The distributions of large droplets and ice particles in P2 were bimodal. The updraft strength in P2 was weaker than that in P1 (Fig. 4c), but P2 was more glaciated than P1 with $F_{Ice}$ values ranging from 0.36 to

210 1 (Fig. 2). P3 and P4 were relatively quiescent compared with the other stages. The cloud-top height in P3 was lower than that in P2, and the area of stronger echoes ($>20$ dBZ) was also smaller than that in P2 (Fig. 4a). Similar to P2, the dissipating stage was dominated by ice, but only intermittent unglaciated LWC-rich clouds were present (Fig. 4d, e); however, the clouds in P3 had a greater glaciated fraction (Fig. 2). These findings confirm that P3 corresponds to the dissipating stage. P4 was likely a newly developed cell with a lower $F_{Ice}$ and weak radar reflectivity, and the cloud-top was not as high as that in

the other stages (Figs. 2 and 4a). This stage was rich in liquid water with an LWC of up to 0.27 g m$^{-3}$ at a colder temperature (−11°C), whereas the IWC measured in the region was significantly lower than those measured in other stages (Figs. 4d, e and 5).

Figure 5 summarizes the relationships between LWC and IWC at different stages of cloud development. For the newly developed cell (P4), the high LWC with less IWC ($< 0.2$ g m$^{-3}$) was predominant, and this feature was also present in the

220 developing stage. The other stages with appreciable IWCs corresponded to LWC values less than 0.2 g m$^{-3}$, indicating that the clouds experienced different extents of glaciation. The clouds in P2 were primarily composed of ice water, and the number concentration of cloud droplets was significantly lower than that in P1. P3 was identified as dissipating cells when the clouds were dominated by ice water and had a higher $F_{Ice}$ than in P2 (Fig. 2).

## 3.2 Ice production at different stages of cloud development

Figure 6 shows the vertical profiles of microphysical properties at different stages. Figures 6a1-a4 and 6c1-c4 are coloured by the effective diameter of the droplets and $F_{smaller\ ice}$, respectively. Several targeting periods of P1, P2, and P4 were selected for detailed analysis, including periods 1.1, 1.2, 2.1, 2.2, 2.3, 4.1, 4.2, and 4.3 (the specific times are given in Table S1), and the corresponding periods are marked in the time series of Fig. 4. In developing cells, with increasing height, $N_{FCDP}$ tended to decrease, whereas the diameter of the droplets tended to increase (Fig. 6a1), and there was an increase in $N_{Round}$ at two levels (Fig. 6b1). A broadened droplet spectrum at two levels of developing cells was also observed (Fig. S4). Period 1.1 (abbreviated P1.1, the same for other periods) corresponded to high $N_{Ice}$ with less LWC, and P1.2 corresponded to the region with less ice and some LWC (Fig. 6c1). The size spectrum in Fig. 7a shows that the $N_{FCDP}$ and $F_{smaller\ ice}$ at P1.2 were both greater than those at P1.1, and precipitation particles had formed at P1.1, whereas P1.2 was still dominated by smaller droplets with few precipitation particles (Fig. 6d1). Clear similarities were observed between the two periods: $N_{round}$ in both periods was greater than that in the other unmarked periods in P1, and the average $N_{Round}$ exceeded 30 L$^{-1}$ (Fig. 7a), with a maximum $N_{Round}$ greater than 50 L$^{-1}$ (Fig. 6b1). In addition, the larger size determined by the 2D-S than the FCDP was found in Fig. 7, which is due to the lower accuracy of the 2D-S in determining the particles in smaller bins (Gurganus and Lawson, 2018; Woods et al., 2018). This may be particularly the case when some small non-spherical ice particles are present at colder temperatures.

P1.1 and P1.2 showed $N_{Ice}$ up to 256 L$^{-1}$ and 71 L$^{-1}$, respectively. Considering the factor of 10, which is the uncertainty pointed out by Demott et al. (2010), the observed ice concentration was still approximately 2 orders of magnitude higher than the calculated INP in the corresponding temperature regime (Fig. S5). The ice shapes were dominated by the plate, irregular and linear ice categories (Fig. 7a), and the 2D-S images revealed H-shaped ice crystals, with the ice particles exhibiting obvious riming characteristics. The ice habits were consistent with the features of cloud regions in which SIP is thought to be active (Field et al., 2016). Considering that the temperature of the environment was within the H–M zone and that the region was rich in supercooled large droplets, the H–M process was most likely active (Crosier et al., 2013; Taylor et al., 2016). The ice production in P1.1 and P1.2 appeared to be triggered by the riming process of large ice particles, and the temperatures of the two periods also indicated the likely H–M process for SIP during this stage (Fig. 7a). The difference between the two periods was that P1.1 seemed to have completed the SIP process and formed precipitation particles, whereas there were still many cloud droplets in P1.2 with fewer large ice particles. This might suggest that the large number of large ice particles in P1.1 improved the riming efficiency and increased the riming surface area, leading to more small ice particles through the H–M process and resulting in the consumption of the droplets. However, the dynamic vertical or horizontal transported of ice, e.g., in convective thermals, the ice near the cloud-top can be circulated downwards surrounding the convection core, while being transported upwards in the convection core (Korolev et al., 2020). This might induce some uncertainty when evaluating the concentration at the aircraft observed position.

The cloud-top height reached 10.1 km in mature cells, and the temperature at this stage was lower than the H–M temperature regime. P2.1, P2.2 and P2.3 corresponded to areas with high, modest, and low concentrations of ice in P2, respectively. $N_{Round}$ decreased gradually from P2.1 to P2.3, and P2.2 had more cloud droplets (Fig. 6a2-d2). The size spectrum in Fig. 7b shows that the $N_{Ice}$, $N_{Round}$ and $F_{smaller ice}$ values in P2.1 were all greater than those in P1.1. Plate, irregular, and linear ice also accounted for the majority of the ice in P2.1, and the riming characteristic of large ice in P2.1 was clearly shown in the images (Fig. 7b). Although the average temperature of P2.1 was as low as -11.7 °C, the abundant large ice particles seemed to trigger the active SIP process in P2.1, with a high $N_{Ice}$ of approximately 300 L$^{-1}$. This finding indicates that the SIP process might not be restricted by temperature, although the possible transport of ice from other cloud regions cannot be completely excluded. The period 2.2, which lacked enough large ice, was likely still in the glaciation process, and P2.3 might have had difficulty triggering a more active SIP process due to the smaller number of large ice particles and limited liquid water. Notably, the observed $N_{Ice}$ may have involved hydrometeors transported from other parts of clouds, along with the locally produced ice. Ice production can therefore be considered a continuous process, and the observed $N_{Ice}$ is a net production of ice after considering all the input (local production and transport in) and output (fall out and transport out) factors at the observation level.

In dissipating cells, the clouds were dominated by ice, and $F_{smaller ice}$ decreased, indicating that the ice production process was completed (Fig. 6a3-d3). The clouds in P4 were dominated by liquid water and classified as young cells, with a cloud-top at only 5.5 km (Fig. 6a4-d4). The vertical profiles revealed that P4.1 and P4.3 were dominated by droplets with few ice particles and large droplets, whereas P4.2 featured large droplets with few droplets. The ice particles observed at this stage most likely originated from the ice nucleation process and ice falling from above. The aircraft penetrated the cloud-top in P4.3, and several ice particles (Fig. 7c), which were likely primary ice particles, were observed. The size spectrum and 2D-S images in Fig. 7c show that large ice particles were present in P4.1, and the images suggest that these particles were likely formed through riming and Bergeron processes, whereas the ice in P4.2 was mainly smaller ice, possibly still in the process of growth.

The large ice particles falling from the upper level likely played a very important role in the ice production process, where the primary ice crystals might have formed through the nucleation process and grew in the upper level or during the fall, then fall to the lower level to trigger the ice production process. However, the number of large ice particles was not the only factor determining the ice production process; large droplets also played a significant role in promoting the SIP process. Figure 8 shows scatter plots of the corresponding distributions of $N_{Ice}$ and $N_{Round}$ at different stages, coloured by the diameter of the large droplets. There was a positive correlation between $N_{Ice}$ and $N_{Round}$, with more large droplets generally corresponding to a higher $N_{Ice}$. A comparison of P1.1 and P1.2 reveals that larger large droplets tended to produce greater $N_{Ice}$ at the same $N_{Round}$. The large droplet with a diameter of 160 μm corresponded to almost 5-folds ice numbers of that of 80 μm, and Figure 8 also clearly shows the importance of the larger large droplet in the production of more ice particles in P2. On the basis of the above analysis, when a high number of large ice particles fell from the upper level to the lower level, if

there were abundant larger large droplets in the lower level, the riming efficiency could be improved, and the SIP process could be enhanced.

### 3.3 Ice production determined by the distance to cloud-top

Figure 4 shows that even at the same level, $N_{Ice}$ varied by two orders of magnitude from less than one to a few hundred per litre. This means that during aircraft penetration, different intensities of SIP events were experienced. The primary cause of this variability was attributed to the position of the aircraft relative to the cloud-top, i.e., the distance to cloud-top (DCT) during the measurement.

Figure 4b shows the time series of DCT during the experiment. When penetrating a cloud turret, the aircraft entered the cloud with a low DCT, reached a higher DCT when close to the convective core, and left the cloud with a low DCT again. The results therefore revealed a few humps of DCT values during a few penetrations of convective cells or more spread parts of the clouds. The DCTs ranged from 0.01 to 4.6 km during the experiment. Figures 4b and d show that the higher DCTs (2.8 and 4 km, respectively) corresponded to the peak values of $N_{Ice}$ (256 and 300 $L^{-1}$, respectively) in P1 and P2. For each penetration, $N_{Ice}$ increased dramatically when the aircraft was closer to the cloud core with a higher DCT and decreased upon leaving. This clearly indicated a positive correlation between DCT and $N_{Ice}$.

Figure 9 shows $N_{Ice}$ and $N_{FCDP}$ as functions of DCT for different stages of clouds. In the developing stage, $N_{Ice}$ significantly increased when the DCT was above 2 km, and was positively correlated with $N_{Ice}$ up to DCT of 3 km. For the mature and dissipating stages, $N_{Ice}$ increased from the cloud-top (DCT = 0.2 km) to a certain DCT but decreased with increasing DCT. This suggested that the development of the cloud-top increased $N_{Ice}$, and considering that larger particles tended to fall to the cloud base and form precipitation, the reduced $N_{Ice}$ close to the cloud base may be due to the coalescence of ice, which reduced the number but increased the size of the ice. It should be noted that the observed clouds have included both widespread stratiform and embedded convective clouds, and the DCT metric should apply to all these clouds. The DCT essentially implies that the amount of ice hydrometeors may fall from above, but may not be directly associated with the current updraft strength or turbulence.

$N_{Ice}$ could increase from a few dozen to a few hundred per litre, which are all well above the estimate from INP, indicating strong SIP. For the SIP mechanism, the temperature of P1 (−5 to −8 °C) was in the typical H–M temperature region, whereas the temperature of P2 (−12 °C) was lower than the H–M temperature region. Even at the same ambient temperature in the measurements (because the aircraft penetration was at the same altitude), the $N_{Ice}$ showed a marked difference. This suggested that the DCT played an important role in the SIP process, and in regions with temperatures lower than the H–M temperature zone, the DCT tended to be a more important factor than temperature in determining the intensity of SIP.

### 3.4 The production rate of secondary ice

The secondary ice production rate can be estimated through the measured number size distribution of ice (Harris-Hobbs and Cooper, 1987; Crosier et al., 2011). The concentration between the lengths of 90–140 μm ($N_{90-140μm}$) was divided by the time

required for ice to grow under this size range. The ice grew linearly under water supersaturation within this size range and was approximately 1.4 μm/s at T = −6°C (Ryan et al., 1976), resulting in around 35.7 s to grow for 50 μm ($\tau$). It was assumed here that the ice numbers were in a steady state such that the smaller ice at size ($L$) = 90 μm grew to $L$ = 140 μm

was replenished by smaller ice newly produced purely by splinters. The production rate of the smaller secondary ice could then be estimated by the ice number between this growth size limit ($N_{90-140nm}$) divided by the time required for growth ($\tau$). Figure S6a shows that the measured SIP rate ranged from 0.005–1.8 $L^{-1}$ $s^{-1}$, which is generally consistent with previous observations of 0.001–1 $L^{-1}$ $s^{-1}$ for cumulus clouds (Harris-Hobbs and Cooper, 1987), 0.043 $L^{-1}$ $s^{-1}$ for stratus cloud embedded with cumulus (Crosier et al., 2011), and 0.14 $L^{-1}$ $s^{-1}$ in the mature region of cumulus (Taylor et al., 2016). On the

basis of the observation data of mixed-phase stratiform cloud systems over northern China, Hou et al. (2021) estimated the SIP rate and reported that the highest concentration of ice splinters could reach 1000 $L^{-1}$ in five minutes, which implied that the average SIP rate could reach 3.3 $L^{-1}$ $s^{-1}$. Figures S6c and S7 show that the rate was positively correlated with the number concentration of large ice (graupel) and large droplets.

The above analysis revealed the importance of the collision–coalescence process in producing the enhancement of ice

number concentration. The collision-coalescence model has previously been used to calculate the production rate of secondary ice. It is essentially determined by the collision-coalescence between graupel and droplets above a certain size. It was long established in the laboratory that only droplets > 25 μm in diameter can produce secondary ice when rimed on graupel. The SIP rate can therefore be calculated from the collision-coalescence process between graupel and droplets (Reisner et al., 1998), and the calculation equation is as follows:

$P = \pi/4 \cdot (D_{graupel} + D_{droplet})^2 N(D_{graupel})N(D_{droplet})E|U_{graupel}-U_{droplet}|,$ (3)

where $D_{graupel}$ and $D_{droplet}$ are the effective diameters (which are the third divided by the second moment of the size distribution) of the graupel and droplets, respectively; $N(D_{graupel})$ and $N(D_{droplet})$ are the number concentrations of graupel and droplets, respectively; and $U_{graupel}$ and $U_{droplet}$ represent the terminal velocities, which are calculated as the absolute difference between the graupel and droplets, where $U_{graupel} = 7 \times 10^2 D_{graupel}$ and $U_{droplet} = 3 \times 10^7 D_{droplet}$. $E$ is the collection efficiency

among the size bins of graupel and droplets, which was assumed to be 1 for the first instance but would be discussed as follows. Ice particles with $d$ > 250 μm were considered graupel and were able to capture droplets efficiently (Harris-Hobbs and Cooper, 1987). Here, the effective radius (Re) was used to represent the size distribution of graupel/droplets within a time window to simplify the calculation of collisions among size bins. The Re was used rather than the median mass value from the size distribution because the former was determined by the cross section of the particles (and collection by the

collision-coalescence process was also determined by area) and weighted towards larger particles. Ice particles were observed to be mostly rimed in the images; thus, all the ice particles with $d$ > 250 μm were considered graupel particles that had already accreted small droplets (i.e., $d$ < 13 μm), but the fraction of rimed surface was not calculated (Harris-Hobbs and Cooper, 1987). Considering that the observation here was actually after the SIP process was initialized, when the smaller

cloud droplets had been considerably consumed and most ice particles were rimed, the number of large droplets ($d > 50$ μm) was the limiting factor for SIP and was therefore used to calculate the modelled SIP rate.

Figure S6a shows the time series of the modelled SIP rate, which was well correlated with the measured SIP (the correlation coefficient was 0.86), and the ratio between the Re values of large droplets and graupel ($Re_{Round}/Re_{Graupel}$) ranged from 0.1-0.8 (Fig. S6b). Figure 10 shows the correlation between the measured and modelled SIP rates, coloured by $Re_{Round}/Re_{Graupel}$. According to Eq. (3), the collection efficiency $E = 1$ was first considered, which gave the upper limit for the calculation, but any other circumstances would cause $E < 1$ and reduce the model results. The model was close to the observation when $Re_{Round}/Re_{Graupel}$ ranged from 0.4-1 (slope=0.94), but the model started to overestimate compared to the observation when $Re_{Round}/Re_{Graupel}$ decreased (shown by the data points grouped as different levels of $Re_{Round}/Re_{Graupel}$). This clearly indicates a decrease in $E$ when $Re_{Round}/Re_{Graupel}$ decreased. $E$ was then further adjusted to obtain the modelled SIP rate matching the observations at different levels of $Re_{Round}/Re_{Graupel}$, as shown in the subplot of Fig. 10. A linearly increasing collection efficiency was found, when $E$ increased from 0.2 to 1 as $Re_{Round}/Re_{Graupel}$ increased from 0.1 to 0.7. This was consistent with the theory of droplet collision; when the collector particle approaches the droplet, the droplet tends to follow the streamline around the collector particle and may avoid collision (Wallace and Hobbs, 2006; Pruppacher and Klett, 2010). The collision efficiency was low when the collector particle was much larger than the droplet because overly small particles would follow the streamline around the collector particle due to their low inertia, and the collision efficiency increased with increasing droplet size because droplets with greater inertia tended to follow a straight line. The results here imply that the SIP rate can be well explained by the collision theory between graupel and large droplets, and the availability of both numbers and the chance for their collision were the factors determining the SIP rate.

## 4 Discussion and conclusions

In this study, we investigated the ice production in stratiform clouds with embedded convection during an extratropical cyclone over the North China Plain through in situ measurements of microphysical properties. The aircraft penetrated clouds corresponding to four stages of the cloud lifecycle, including developing, mature, dissipating and young cells. The four relative stages were identified by the ice mass fraction, considering that a more mature cloud has a greater glaciated fraction. In developing cells, high-$N_{Ice}$ and LWC-rich regions were observed, and the ice mass fraction in these clouds spanned from zero (pure water) to one (pure ice). In mature cells, a greater extent of glaciation was observed, with the ice mass fraction ranging from 0.36 to 1 and $N_{Ice}$ reaching 300 L$^{-1}$ in this stage. The dissipating cells were dominated by ice but only intermittent unglaciated LWC-rich clouds, and the ice mass fraction ranged from 0.56-1. The young stage was rich in LWC and had a lower ice mass fraction. The $N_{Ice}$ frequently well exceeded that from ice nucleation, reaching up to a few hundred per litre, indicating strong SIP.

The results revealed generally enhanced SIP when greater distance to cloud-top, which could be explained by the seeder-feeder mechanism occurring in stratiform cloud precipitation (Hobbs and Locatelli, 1978; Hobbs et al., 1980; Matejka et al.,

1980): when the cloud-top is higher, more primary ice particles form at colder temperatures and fall. The ice particles can capture smaller liquid water droplets when falling, during which they can grow and the fall speed can be accelerated. This process can considerably enhance the interaction between ice and water droplets or among ice particles, which is necessary for the occurrence of ice fracturing, thereby leading to the avalanche SIP. The age of ice could be estimated on the basis of the fraction of smaller ice ($F_{smaller\ ice}$) here, with the assumption that recently formed ice particles are smaller in size. This implied the pronounced production of smaller ice particles by SIP processes, with $F_{smaller\ ice}$ reaching 70% during the developing period, whereas a lower $F_{smaller\ ice}$ (0.2–0.6) indicated the growth of ice and smaller ice was consumed during the dissipating stage (Fig. 4). This explanation is also similar to the results reported by Li et al. (2021), who reported that columnar ice crystals were produced at the lower level and were seeded by ice particles falling from the upper level.

The likely schematic plot of ice production at different stages of clouds is given in Fig. 11. A higher cloud-top leads to the formation of more primary ice through the nucleation process, and the ice can grow in the upper level and during the fall. The SIP process is triggered when ice particles in the upper level fall to the lower level with supercooled water, initiating the interactions between ice and droplets. In regions with larger DCTs, ice particles in the upper level have sufficient time and distance to grow larger during the fall, and the fall speed can also be accelerated, resulting in more and larger ice particles falling to the lower level. Consequently, the intensity of the SIP process becomes stronger in this region because the falling large ice particles enhance the interactions between ice and droplets, as well as among ice particles. However, larger ice particles may also fall into the H–M zone in mature cells and trigger the SIP process. Moreover, this possible seeder-feeder process was found to extend the SIP process beyond the slightly supercooled temperature region for the typically considered H–M process. The intensity of SIP was to the first order determined by the numbers of graupel and droplets, because the collision and coalescence processes among these hydrometeors necessitated the fracturing of ice. The modelled and measurement-based calculations showed that appropriately treating the size distribution hereby the determination of collection efficiency will improve the modelling of the SIP rate.

Our results indicate that once the cloud-top reaches a sufficient height, the ice initialized from nucleation may boost the avalanche glaciation process when falling ice reaches lower levels in clouds. It should be noted that whether the falling hydrometeors were the ones generated by the ice production process or were about to participate in the ice production process at the same level, may never be separated due to the short time scale of the collision process. However, this is a continuous process that may involve both already-formed and ongoing-happening particles, and the observed or modelled results are an overall net production of ice. The ice particles falling from aloft increase the number of graupel particles and the chance of collision between graupel and droplets and then trigger the SIP process; therefore, the seeder-feeder and SIP processes may occur simultaneously after the SIP process has initialized. The results concerning the microphysical properties of stratiform clouds with convective cells under different stages suggest that the falling hydrometeors associated with the cloud-top height importantly control the cloud glaciation and precipitation processes, and this information may also help find the region of supercooled water in clouds for weather modification work.

**Data availability**

The data in this study are available from the authors upon request.

**Author contribution**

YD and DL analyzed the data and wrote the manuscript, and this work was completed under the guidance of DL, MH and DD. DeZ, PT, WX, WZ, HH, BP, YJ, JS and FW contributed to the aircraft data processing and analysis. DW, XL and YC
performed the synoptic analysis. DoZ and YH contributed to the radar data processing and analysis. RZ conducted the shape classification of 2D-S images.

**Competing interests**

At least one of the (co-)authors is a member of the editorial board of *Atmospheric Chemistry* and *Physics*. The authors also
have no other competing interests to declare.

**Acknowledgements**

This research was supported by the National Natural Science Foundation of China (Grant Nos. 42205093, 42075084, 42005078).

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

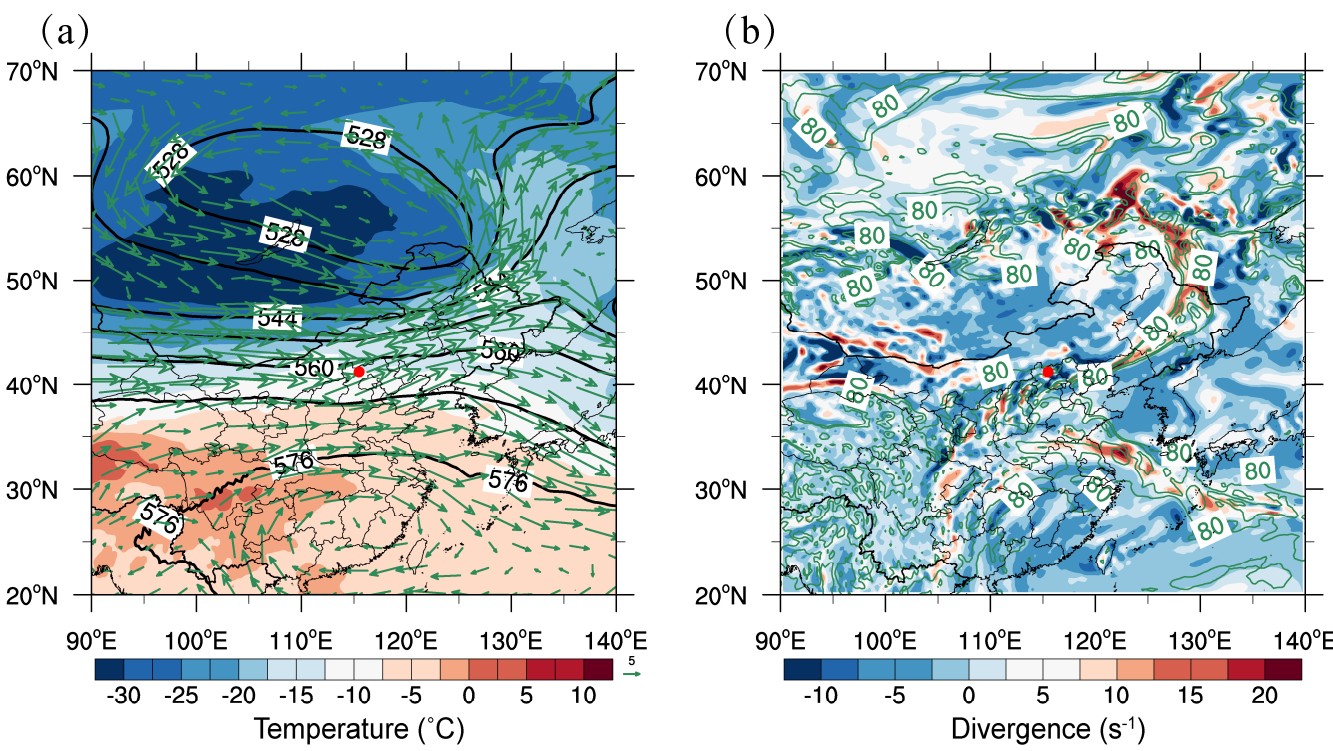

Figure 1: Synoptic overview during the experiment. (a) The 500 hPa temperature (colour), height field (contour), and wind field (arrow) at 08:00 (UTC+8 h) on September 26th, 2017; (b) 850 hPa divergence field (colour) and relative humidity (green line, only >80% is shown). The experimental region is indicated by the red dot on each plot.

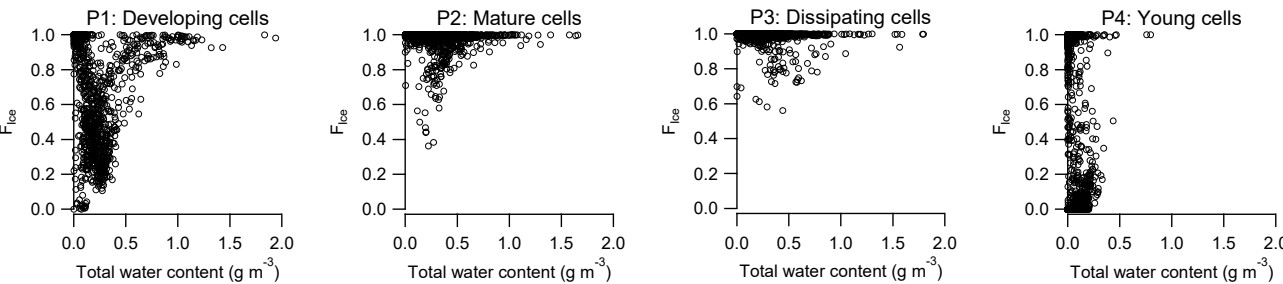

**Figure 2: Ice mass fraction ($F_{Ice}$) as a function of total water content in the four stages.**

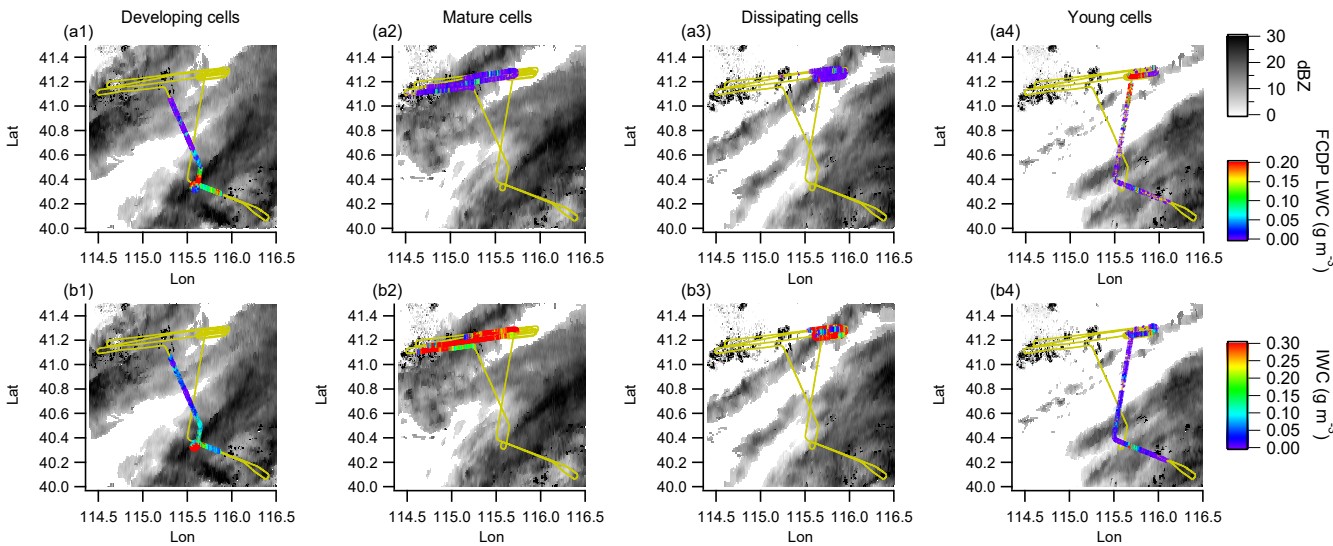

Figure 3: Flight tracks mapping on the composite reflectivity of the S-band precipitation radar at different stages of clouds (from left to right). (a) coloured by the liquid water content (LWC) from the FCDP, (b) coloured by the ice water content (IWC) from the 2D-S.

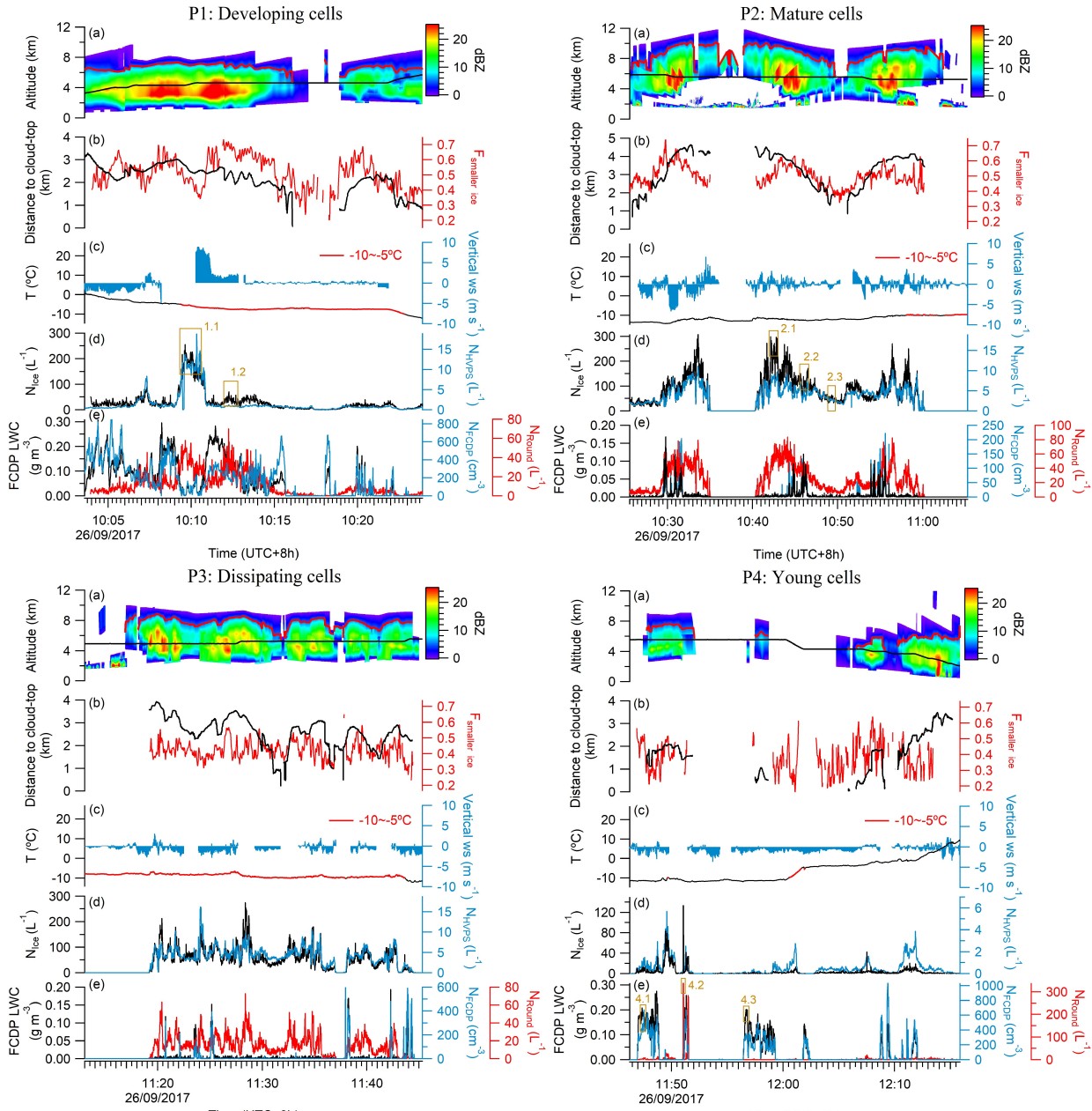

**Figure 4: Cloud properties along the flight track in the four stages. (a) Vertical profile of radar reflectivity from the ground S-band precipitation radar collocated with the flight path. (b)The distance to cloud-top from the aircraft and smaller ice ($d < 180$ μm) number fraction ($F_{smaller\ ice}$). (c) Ambient temperature and vertical wind speed. (d) Ice number concentration ($N_{Ice}$) from the 2D-S and the precipitation particle number concentration ($N_{HVPS}$) from the HVPS. (e) LWC and cloud droplet number concentration ($N_{FCDP}$) from the FCDP and the large droplet number concentration ($N_{Round}$) from the 2D-S. The targeting periods are indexed for further analysis.**

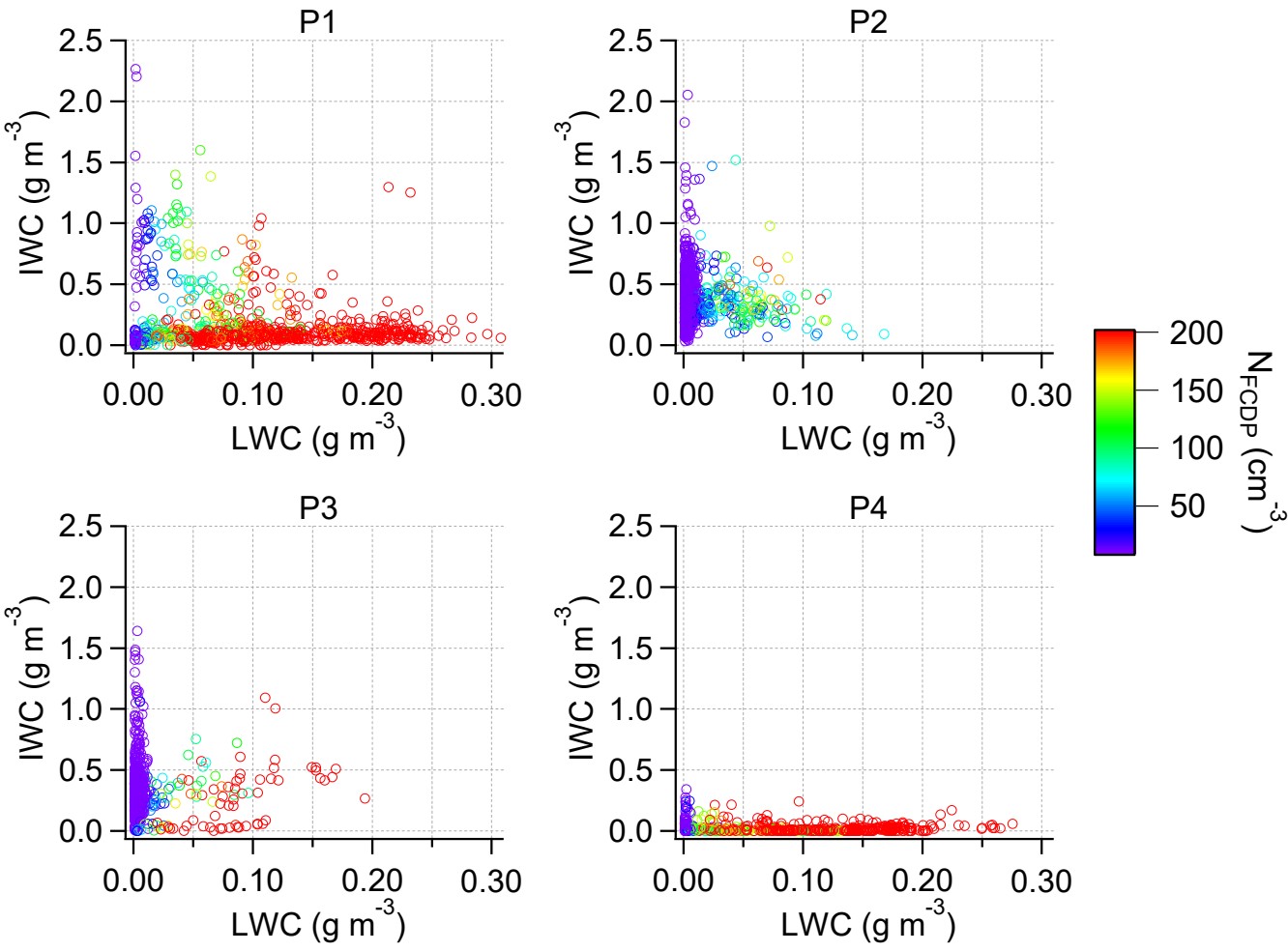

**Figure 5: LWC as a function of IWC at different stages of clouds, coloured by $N_{FCDP}$.**

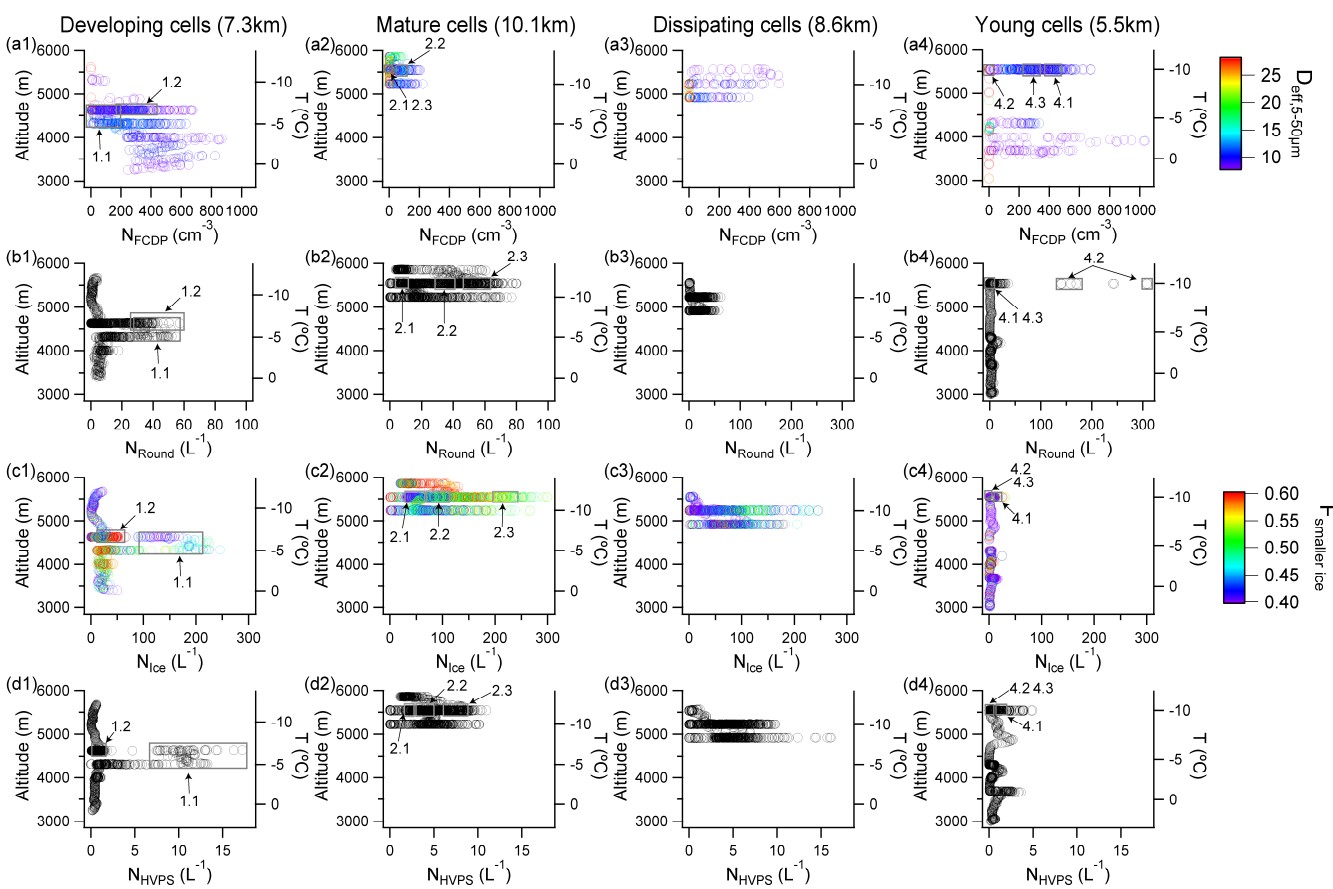

**Figure 6: Vertical distributions of hydrometeors at different stages of clouds. (a)** $N_{FCDP}$ **coloured by the effective diameter of the droplets (5–50 μm), (b)** $N_{Round}$**, (c)** $N_{Ice}$ **coloured by** $F_{smaller\ ice}$**, (d)** $N_{HVPS}$**. The corresponding indexed events in the time series are marked in this figure, and the cloud-top height is indicated in title brackets.**

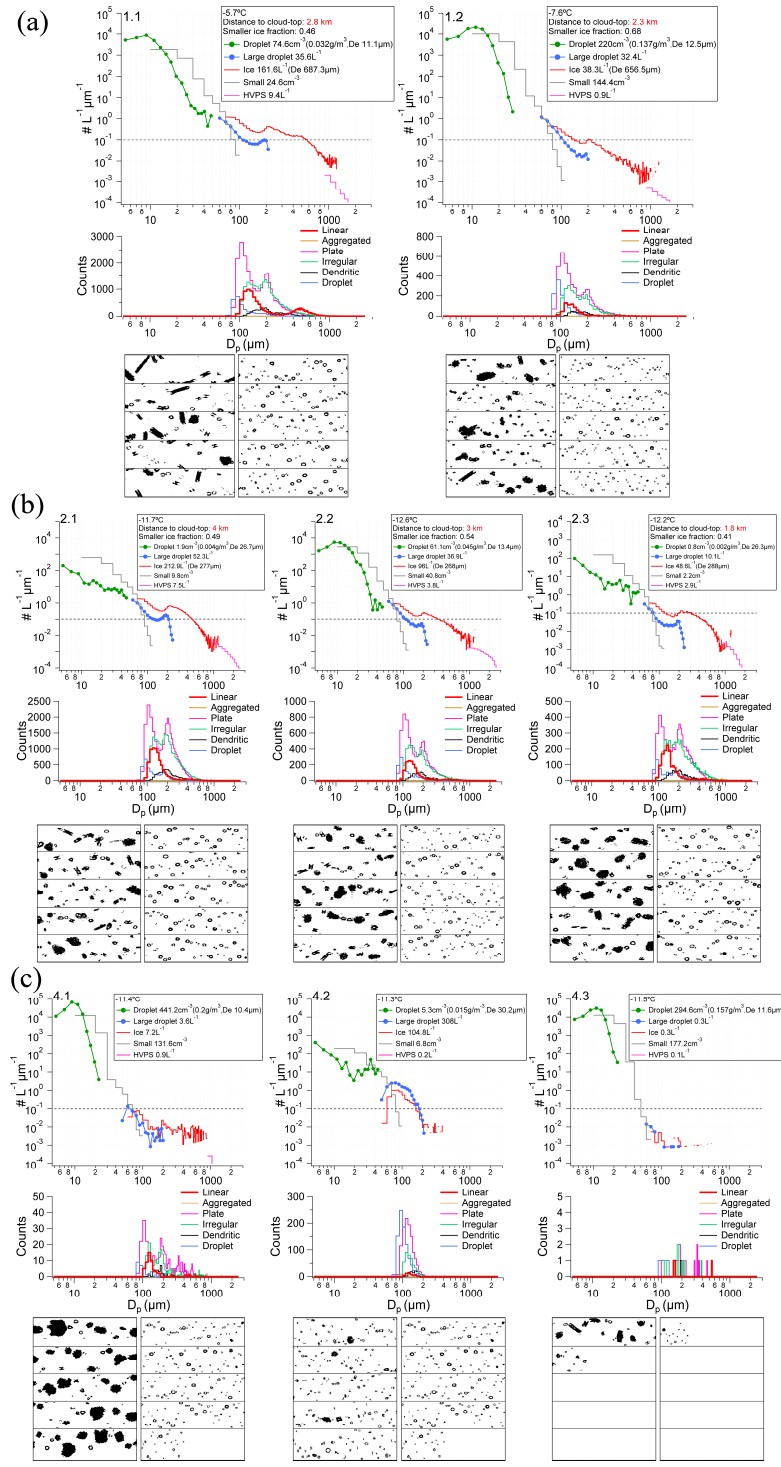

**Figure 7: Particle size spectra from airborne particle spectrum probes, 2D-S images and shape classification results of 2D-S images: (a) periods 1.1 and 1.2; (b) periods 2.1, 2.2 and 2.3; (c) periods 4.1, 4.2 and 4.3.**

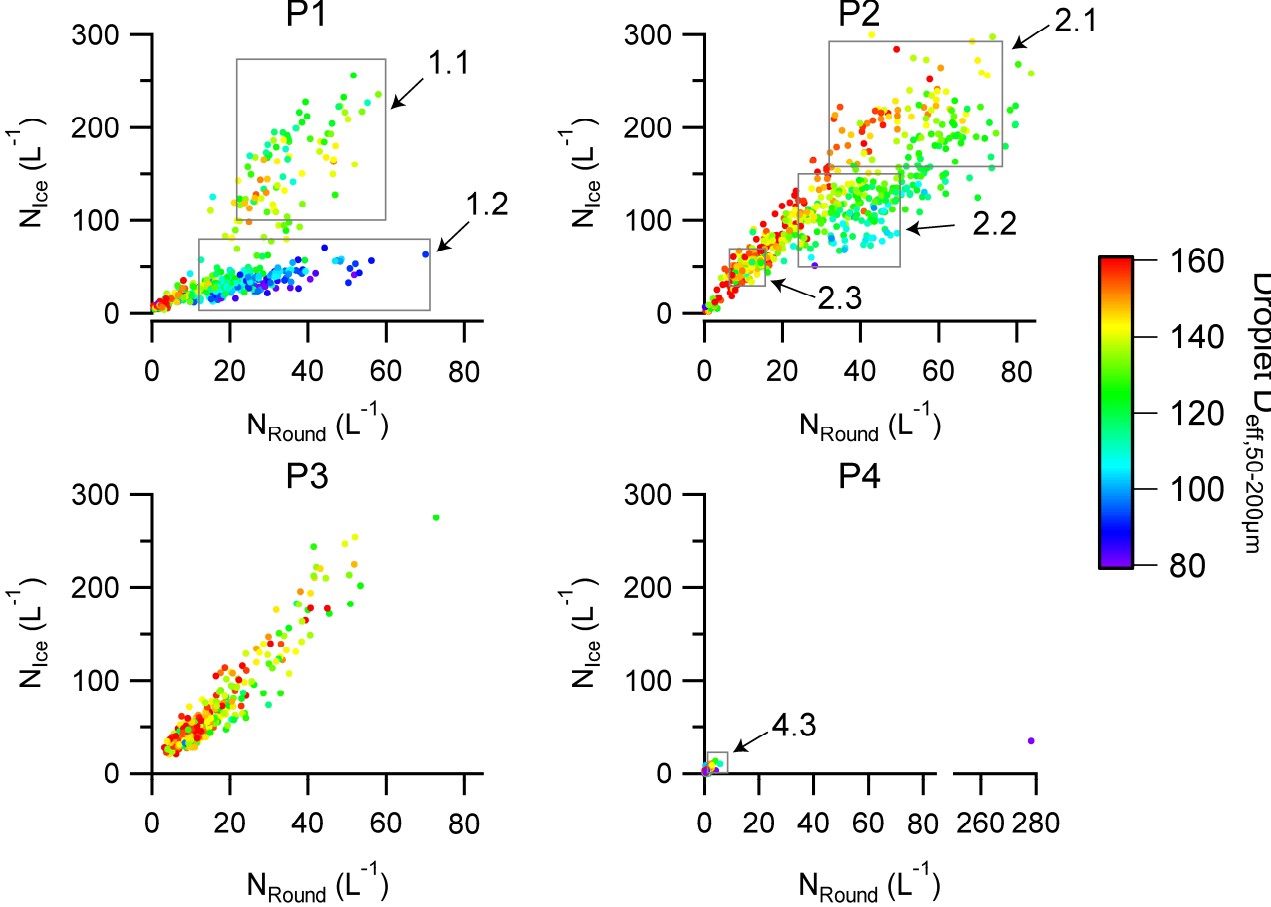

**Figure 8:** $N_{Round}$ **as a function of** $N_{Ice}$ **at different stages, coloured by the diameter of the large droplets (50–200 µm).**

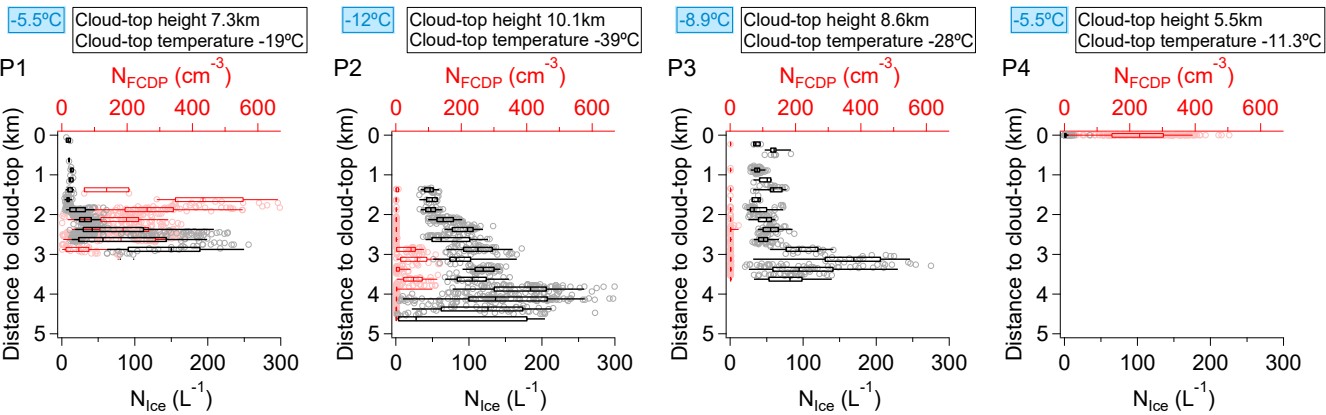

**Figure 9:** $N_{FCDP}$ and $N_{Ice}$ as functions of distance to cloud-top. The grey circled markers and black boxes represent $N_{Ice}$, the light red circled markers and red boxes represent $N_{FCDP}$. Whiskers extend to the 5th and 95th percentiles, boxes encompass the 25th to 75th percentiles, and the 50th percentiles are vertical lines. The blue box in each figure indicates the temperature measured by the aircraft.

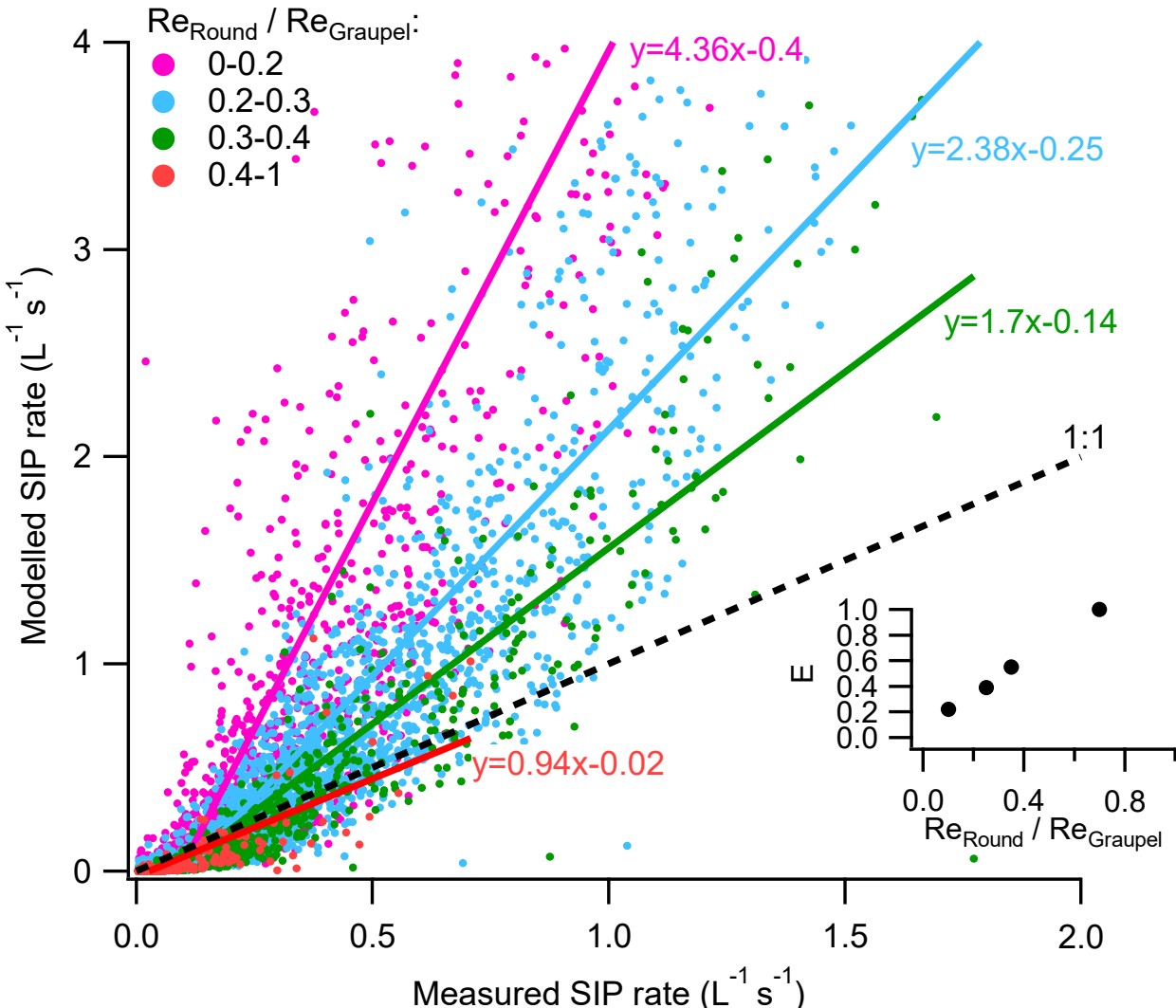

**Figure 10: Measured versus modelled secondary ice production (SIP) rates. The scatter plot is classified and coloured by the ratio of the effective radius between the large droplet and graupel (Re$_{Round}$/Re$_{graupel}$), and each group of data points is subjected to least-square linear fitting. The subplot shows the derived collection efficiency between graupel and large droplets at different values of Re$_{Round}$/Re$_{graupel}$.**

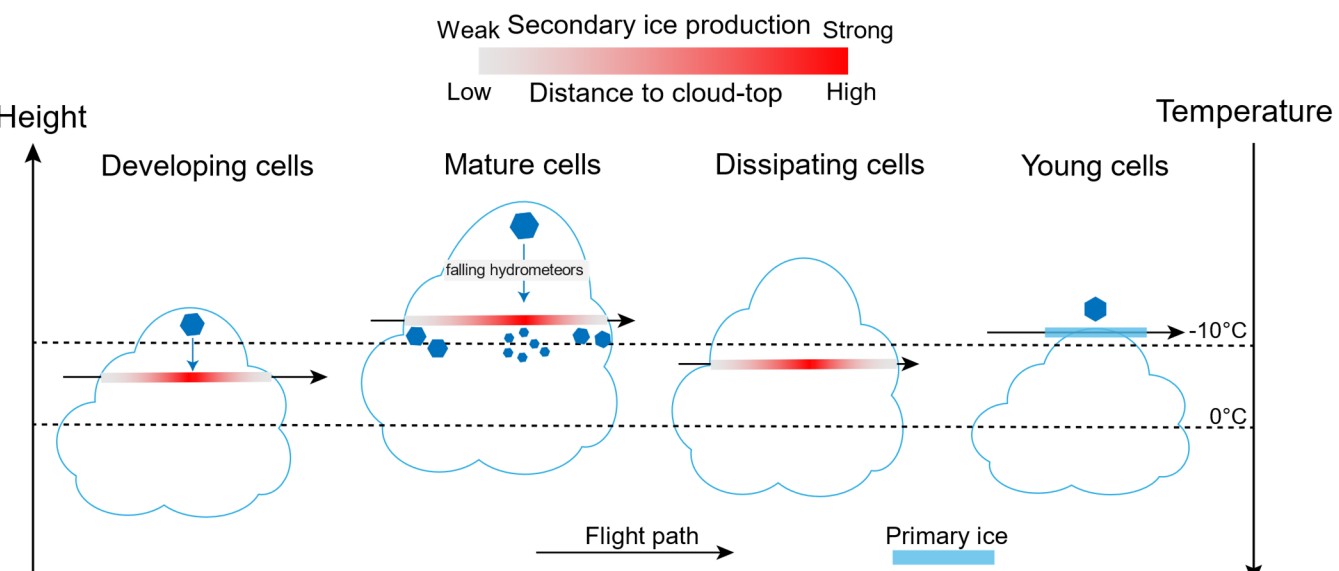

Figure 11: Schematic of ice production at different cloud stages.