# Peer review of "Microphysical view of development and ice production of midlatitude stratiform clouds with embedded convection during an extratropical cyclone"

_EGUsphere, 2024_

## Author Comment (AC1)

We thank reviewers for their important comments to improve our manuscript. We have now addressed all of the comments raised in the revision. The main revisions we made include:

1. We have avoided the discussions about evolution of clouds, and now focus on the discussion and comparison of cloud microphysics at different stages. The ice mass fraction (IWC/TWC) is now used to indicate the different development stages of clouds, by considering a more mature cloud has a more glaciated fraction.
2. We have carefully revised and responded to the related parts which reviewers consider to be speculative.
3. We have added the radar times and the flight time windows for the four stages, and the time windows of targeting analysis periods.
4. We have carefully examined the manuscript for editorial and grammatical errors, addressing them to enhance the overall readability of our work.

In addition, the relative development stages of the clouds are now renamed as P1, P2, P3, and P4 according to the reviewer2's comment. The related descriptions and figures have also been amended in the revised manuscript.

**Reviewer #1**

*It is good to see a set of microphysical measurements being reported. They are difficult measurements to make and are valuable for checking and improving models. Many assertions are made in this paper without evidence. For example it is assumed that there is evolution in time of microphysical characteristics from one cloud region to a different region studied. There is no evidence presented to support this assumption. There are many errors in the written English which need to be corrected. I felt unable to review pp 8-12 of the paper until the comments below are addressed. It is important to state what has been changed on pp 8-12 in response to the above comments if appropriate.*

We thank reviewer for the important comments to improve our manuscript. We now avoid discussing the "evolution" from one cloud to another, but using the ice mass fraction (IWC/TWC) to identify the different development stages of clouds and discuss the differences among them. We have also addressed all of the comments raised in the revision.

**Detailed comments**

*Line 13. Are the clouds stratiform with embedded convection?*

We thank reviewer to point this out, and the cloud type is now changed to "stratiform clouds with embedded convection".
The title is changed as:

Microphysical view of development and ice production of mid-latitude ==stratiform clouds with embedded convection== during an extratropical cyclone.
The related parts are also revised.

*Line 18. Some definitions have graupel only forming when d > 250 um. They are not large at that point.*

We agree with the reviewer's comment, and the related discussions are revised:

P1, Line 19: "The $N_{ice}$ was positively associated with the number concentrations of graupel with diameter ($d$) > 250 μm and large supercooled droplet ($d$ > 50 μm)."

*Line 21. Between what?*

The sentence is more clearly stated:

P1, Line 22: "The collection efficiency between the graupel and droplet was found to increase when the size of droplet was closer to graupel which may improve the agreement between measurement and model."

*Lines 23-26. This is conjecture.*

The related discussions are now revised according to reviewer's following comments.

*Line 80. As above, stratiform with embedded convection? They are not stratocumulus clouds.*

It has been revised in manuscript.

*Line 82. Does the temperature probe become wet in cloud? Most temperature probes do suffer from this problem.*

Thank reviewer to point this out. We found we have used the temperature measured by Rosemont total temperature probe, not the AIMMS probe. We agree with reviewer that the temperature sensor would become wet in cloud and it might underestimate the ambient temperature when water was evaporated, and the Rosemount temperature probe may be affected when T > 0 °C (Heymsfield et al., 1979; Lawson and Cooper, 1990). However, previous literatures suggested this error is negligible in supercooled cloud (Lawson and Rodi, 1992; Korolev and Isaac, 2006), and in this study, the temperature shift in and out of cloud was not observed. The related description has been corrected:

P3, Line 83: "The air temperature was measured by the Rosemount total-air temperature probe (Lenschow and Pennell, 1974; Lawson and Cooper, 1990). The temperature may be underestimated due to water evaporation however this artifact is negligible for supercooled clouds (Lawson and Rodi, 1992; Korolev and Isaac, 2006), and the temperature shift in and out of clouds was not observed in this study."

*Lines 98-99. What is the distance of the radar in Beijing to the cloud system studied?*

The distance from the radar (located in the southeast of Beijing) to the observed cloud system in this study was approximately 50-200 km. This is added in the revision:

P4, Line 103: "==The distance from the radar to the observed cloud system in this study was approximately 50-200 km.=="

***Lines 114-116. What are the errors of the mass of ice determined in this way?***

The error in calculating ice mass according to the mass-dimension relationship from Holroyd (1987) will increase when ice particle is larger and the shape has large irregularity to be classified. More details about the calculation of ice mass from 2DS can be found in Holroyd (1987). The related discussions are added:

P4, Line 121: "==It should be noted that the error in calculating ice mass according to the mass-dimension relationship will increase when ice particle is larger and the shape has large irregularity to be classified (Crosier et al., 2013)==."

***Line 136. It would be helpful to show the location of Beijing in Figure S2. Although the text mentions that Figure S2 shows the movement of the surface cold front, the location of the surface cold and warm air is the same over the 4-hour period for the two flight segments. Is this relevant?***

The location of Beijing is now added in Fig.S2. We have now revised Fig. S2 and Fig. S3 to more clearly shows the surface position of front system according to the wind and temperature shear. The southeastward movement is indicated here (green box) to better indicate the position of aircraft relative to the front system:

[Figure]

Figure S2: Flight tracks (black line) mapping on the wind field (blue wind shaft) and temperature field (color) observed by ground weather observation station at 09:00-12:00 (BJT) on September 26[th], 2017. ==The location of Beijing is indicated by the black five-pointed star.==

*Lines 137-140. Is Figure S3 essential? Perhaps add the location of Outer Manchuria and the surface cold front if the figure is kept.*

Figure S3 shows the location of extratropical cyclone and indicates the aircraft observation area relative to the position of extratropical cyclone. The location of Outer Manchuria and the surface cold front are added:

[Figure]

Figure S3: Geopotential height contour map at 1000 hPa at 09:00 (BJT) on September 26th, 2017. The experimental region and the center of the extratropical cyclone are indicated by the red dot and yellow dot respectively, and the surface cold front is indicated by the blue cold front symbol.

*Lines 143-144. The sentence is not clear. Where was the aircraft observation area?*

This is more clearly stated:

P5, Line 151: "The aircraft sampled the clouds formed in this cyclonic system at this stage, i.e. behind the surface cold front line (Fig. S3) and for the newly formed, developing and matured clouds."

*Lines 151-153. It would be helpful to provide the times of the radar plots and indicate key times of the aircraft tracks to show the development of the system. It looks like the developing cells were observed in the southerly part of the system, whereas the mature cells were observed in the northern part. It is assumed, I think, that the clouds have the same dynamical and microphysical properties in the two regions. It would be good to discuss this point in the paper.*

We thank reviewer to point this out, the times of the radar plots and the start/end time for each stage are added. The clouds observed were in the same cyclonic system, where the clouds underwent continuous generation, development and dissipation. The microphysical properties under different development stages of clouds are therefore different. However, though in the same synoptic system, it is difficult to explicitly rule out the "evolvement" from the same cloud, because we were measuring different clouds. We therefore now avoid the discussions about evolution of clouds, but using the ice mass fraction (IWC/TWC) to indicate the different development stages of clouds (Fig. 2), by considering a more mature cloud has a more glaciated fraction, for the discussions of cloud microphysics at different stages.

P6, Line 164: "The flight tracks mapping on the composite reflectivity of precipitation radar are shown in Fig. 3, colored by the LWC from FCDP and IWC from 2D-S respectively, the radar times and the flight time windows for the four stages are shown in Table S1."

Newly added Table S1 (Supplement):

Table S1. Radar times in Fig. 3 and the corresponding flight time windows for the four stages (all in UTC+8h).

| Stages | Radar time | Flight time window | Targeting periods |
|---|---|---|---|
| Developing cells (P1) | 10:06 | 10:03:35-10:23:55 | 1.1: 10:09:13-10:10:31
1.2: 10:11:54-10:12:41 |
| Mature cells (P2) | 10:42 | 10:25:35-11:05:17 | 2.1: 10:42:02-10:42:58
2.2: 10:45:47-10:46:25
2.3: 10:48:38-10:49:33 |
| Dissipating cells (P3) | 11:30 | 11:13:01-11:45:20 | \ |
| Young cells (P4) | 12:00 | 11:46:16-12:15:52 | 4.1: 11:47:16-11:47:35
4.2: 11:51:02-11:51:05
4.3: 11:56:30-11:56:50 |

Newly added Figure 2:

[Figure]

Figure 2: Ice mass fraction ($F_{Ice}$) as a fuction of total water content at four stages.

Related discussions are added:

P5, Line 156: "Four relative stages during the lifecycle of clouds were identified during experiment, which corresponded to the developing (P1), mature (P2), dissipating (P3) and young cells (P4) in cloud system, according to the different glaciation extents of clouds. The ice mass fraction ($F_{Ice}$: IWC/TWC) was used to indicate the different development stages of clouds (Fig. 2), by considering a more mature cloud has a more glaciated fraction, for the discussions of cloud microphysics at different stages. The cloud system was formed through the combined effects of dynamic forcing induced by the frontal uplift and the moisture transport provided by the prefrontal southerly air mass. Therefore, this study postulated that the continuous clouds within the cloud system had similar dynamic and thermodynamic properties."

***Line 154. I don't think "evolved" is the correct word? Is it the case that the aircraft passed from a region with dominant LWC to one with dominant IWC? It is also curious why the LWC was so low in "developing cells". Is the cloud base altitude known?***

We agree with reviewer that we were sampling different clouds and the word "evolve" is removed in the revision. The developing stage has mixed clouds with a range of LWC from 0.002 to 0.31 g m$^{-3}$ but a lower $F_{Ice}$ than mature stage. The cloud base is at about 1 km according to the radiosonde measurement.
Related discussions are revised:

P6, Line 166: "In developing cells, substantial LWC was detected up to 0.3 g m$^{-3}$, and the aircraft penetrated a high IWC region in this cloud at 10:09-10:11 BJT, with the highest IWC exceeded 2 g m$^{-3}$ (Fig. 3a1, b1), and $F_{Ice}$ at this stage could span from zero (pure water) to unit (pure ice) (Fig. 2)."

***Lines 154-155. There is no evidence given that the LWC was "consumed by the growth of ice crystals". Is it the case, from the evidence in Fig 2, that the aircraft made passes through a different region of cloud (at a later time according to Figure 3), and measured higher values of IWC than LWC?***

We agree with reviewer that it is difficult to link the clouds from both areas. We have removed this discussion about the evolution of the microphysical processes. The related discussions are revised:

P6, Line 168: "In mature cells, $F_{Ice}$ ranged in 0.36-1 and IWC generally exceeded 0.3 g m$^{-3}$ (Figs. 2 and 3b2), and the maximum radar reflectivity of mature cells was enhanced from 20 dBZ to 27 dBZ at 10:06 to 10:42 BJT (Fig. 3a1, a2)."

***Lines 155-156. It isn't clear from Fig 2 that the radar reflectivity was enhanced. Is this enhanced from the same region in Fig 2a1 and 2a2? The values of radar reflectivity would be helpful.***

The radar reflectivity of the clouds in mature cells is enhanced compared to the last, and the maximum radar reflectivity increased from 20 dBZ to 27 dBZ at 10:06 to 10:42 BJT. The related discussions are revised:

P6, Line 168: "In mature cells, $F_{Ice}$ ranged in 0.36-1 and IWC generally exceeded 0.3 g m$^{-3}$ (Figs. 2 and 3b2), and the maximum radar reflectivity of mature cells was enhanced from 20 dBZ to 27 dBZ at 10:06 to 10:42 BJT (Fig. 3a1, a2)."

*Line 156. How is it known, From Fig 2 that "the ice phase precipitation process occurred"?*

The clouds moved eastward over time and the aircraft conducted continuous observations of the cloud. The statement that precipitation has occurred is based on the significant weakening of the radar echoes in dissipating cells compared to the mature cells (Fig. 3 and Fig. 4), indicating a reduction in large particles within the cloud.

*Line 159. Figure 3 contains a lot of detailed information. It would be helpful to include shorter time series with the details expanded. For example, what is the structure of the region with a strong downdraft and updraft. It isn't clear where the peaks in LWC and concentration of ice particles occur relative to the updrafts and downdrafts. Also, is the strength of the downdraft real? Have the vertical winds been filtered for aircraft turns? It might be helpful to add vertical dotted lines in Fig 3.*

According to reviewer's suggestion, Fig. 3 (Fig. 4 in revised manuscript) is now broken into a few figures with shorter time window showing finer details, as Fig. S4. The vertical wind data during aircraft turns are removed in Fig. 4 and Fig. S4. The AIMMS-20 probe equipped on the aircraft of this experiment is calibrated once every two years according to the procedure and flight scheme provided by the manufacturer, therefore the vertical wind data observed during level flight of aircraft is reliable. The vertical wind data during aircraft turns are not used for data analysis.
Related description is added:

P6, Line 174: "Figure 4 shows the microphysical properties of clouds and meteorological parameters along the flight track, and Figure 4 is broken into four figures according to four stages with shorter time window showing finer details (Fig. S4). The vertical wind data during aircraft turns were excluded from Figs. 4 and S4 and were not used for the analysis."

Newly added Figure S4:

[Figure]

Figure S4: Cloud properties along the flight track at four stages. (a) Vertical profile of radar reflectivity from the ground S-band precipitation radar collocating with the flight path. (b)The distance to cloud-top of aircraft and smaller ice ($d < 180$ μm) number fraction ($F_{smaller\ ice}$). (c) Ambient temperature and vertical wind speed. (d) Ice number concentration ($N_{Ice}$) from the 2D-S and precipitation particle number concentration ($N_{HVPS}$) from the HVPS. (e) LWC and cloud droplet number concentration ($N_{FCDP}$) from the FCDP, and the large droplet number concentration ($N_{Round}$) from the 2D-S. The targeting periods are indexed for further analysis.

*Line 160. Is "Developing cells" a relative term since the same strength of radar echoes appear as in the mature cells, and the clouds are already quite deep with the top of the radar echoes at about 8 km. Are there radar echoes from earlier and later times of the cells penetrated in S1 just after 10 local?*

We agree with reviewer that the developing cells are relative term. The ice mass fraction ($F_{Ice}$: IWC/TWC) is used to indicate the different relative development stages of clouds,

by considering a more mature cloud has a more glaciated fraction, for the discussions of cloud microphysics at different stages. The developing stage is determined as the $F_{Ice}$ is in the range of 0-1, while the $F_{Ice}$ is in the range of 0.36-1 in mature cells (Fig. 2). The related discussions are revised:

P5, Line 156: "Four relative stages during the lifecycle of clouds were identified during experiment, which corresponded to the developing (P1), mature (P2), dissipating (P3) and young cells (P4) in cloud system, according to the different glaciation extents of clouds."

P6, Line 166: "In developing cells, substantial LWC was detected up to 0.3 g m$^{-3}$, and the aircraft penetrated a high IWC region in this cloud at 10:09-10:11 BJT, with the highest IWC exceeded 2 g m$^{-3}$ (Fig. 3a1, b1), and $F_{Ice}$ at this stage could span from zero (pure water) to unit (pure ice) (Fig. 2). In mature cells, $F_{Ice}$ ranged in 0.36-1 and IWC generally exceeded 0.3 g m$^{-3}$ (Figs. 2 and 3b2), and the maximum radar reflectivity of mature cells was enhanced from 20 dBZ to 27 dBZ at 10:06 to 10:42 BJT (Fig. 3a1, a2)."

P12, Line 383: "The four relative stages were identified by ice mass fraction, by considering a more mature cloud has a more glaciated fraction."

***Line 170. Isn't it the case the S1 and S2 were in different cloud regions? Is it correct to say "developed"?***

We agree with the referee's comment, and the P1 and P2 are different cloud regions. It has been corrected:

P7, Line 197: "The cloud-top height in P2 reached 10 km (Fig. 4a), which was the highest cloud-top of clouds observed during experiment."

***Line 172. There are still a few cores with reflectivity values close to 30 dBZ.***

The area of radar reflectivity exceeding 20 dBZ in dissipating cells significantly decreases compared to mature cells. The dissipating cells is identified due to the clouds have a more glaciated fraction compared to mature cells. The related discussions are added:

P6, Line 170: "At the dissipating stage, the ice phase precipitation process occurred and the radar reflectivity became weaker with narrowed cloud band (Fig. 3a3, b3), and the range of $F_{Ice}$ reached up to 0.56-1 (Fig. 2)."

P7, Line 201: "The cloud-top height in P3 was lower than P2, and the area of stronger echoes (>20 dBZ) was also reduced compared to P2 (Fig. 4a). Similar to P2, the dissipating stage was dominated by ice but only intermittent unglaciated LWC-rich clouds were present (Fig. 4d, e), however, the clouds in P3 had a more glaciated fraction (Fig. 2). All above confirmed the dissipating stage of P3."

*Line 173. It is very surprising that the LWC values are so low with vertical winds of +10 m/s. Is it actually the case that the cloud has suffered significant entrainment and conversion to precipitation?*

Previous studies also observed low LWC when high updraft when liquid water was considerably consumed by rapid production of ice, such as LWC as low as 0.1 g m$^{-3}$ was observed when updraft up to 8.5 m/s (Lawson et al., 2015). The related discussion has been revised:

P6, Line 190: "P1 was strongly turbulent with vertical wind speed up to 8.9 m/s, and the strong updraft region was dominated by ice particles and precipitation particles (Fig. 4c-e). The low LWC when high vertical updraft may be caused by the rapid production of ice particles, which was also observed in tropical highly convective region (Lawson et al., 2015)."

*Lines 174-175. There is no evidence that the liquid was consumed by producing ice. Is it possible that the downdraft is a region affected significantly by entrainment and the ice particles were transported down from above?*

We agree with reviewer and the statement about ice consumption is now removed. The downdraft data is observed during aircraft turns and has been removed now (Fig. 4 and Fig. S4), and the related discussion are revised:

P6, Line 192: "The ice number peaked at a valley between two peaks of liquid water, but it was difficult to determine the vertical wind at the peak of ice number due to aircraft turns (Fig. S4). However, in the subsequent level flight, high ice number concentration (> 170 L$^{-1}$) was also observed in the strong updraft region. After the high ice number region, LWC up to 0.28 g m$^{-3}$ was observed in the region with weaker updraft."

*Line 177. There is no evidence that the drops in S1 grew to larger drops and were consumed by ice in S2.*

It has been revised:

P7, Line 198: "The LWC in P2 was considerably lower compared to P1, while there were more large droplets and ice particles in clouds (Fig. 4d, e), and the distribution of large droplets and ice particles in P2 showed a bimodal distribution."

*Lines 178-179. Is the cloud measured in S4 really stratocumulus cloud?*

The stratocumulus (it has been corrected as "stratiform clouds with embedded convection") is an overall description of the cloud system observed in this experiment. The continuous clouds within the cloud system are postulated to have similar dynamic and thermodynamic properties in this study, therefore the clouds are now described as stratiform clouds with embedded convection.

*Line 181. Some of the cells at the beginning of S3 do not appear to be dissipating. Cloud tops are still above 8 km and the radar echoes are approaching 30 dBZ (which is not a high value, but similar to the values in S2).*

The dissipating is a relative stage, which is identified to have a more glaciated fraction compared to mature cells. Additionally, the area of stronger echoes in dissipating cells is reduce compared to mature cells, and the cloud-top height is also lower. Related discussions are revised:

P7, Line 201: "The cloud-top height in P3 was lower than P2, and the area of stronger echoes (>20 dBZ) was also reduced compared to P2 (Fig. 4a). Similar to P2, the dissipating stage was dominated by ice but only intermittent unglaciated LWC-rich clouds were present (Fig. 4d, e), however, the clouds in P3 had a more glaciated fraction (Fig. 2). All above confirmed the dissipating stage of P3."

*Line 181. S4 includes the first region of (weaker) radar echo with a top above 6 km. Why is it treated as a young cell?*

The clouds are identified as young cells due to the lower glaciated fraction compared to other stages (Fig. 2). The first region in P4 is rich in LWC, and the overall ice mass fraction is smaller than developing cells. This is now added in the revision:

P7, Line 204: "P4 was likely a newly developed cell with lower $F_{Ice}$ and weak radar reflectivity, and the cloud-top had not reached as high as other stages (Figs. 2 and 4a). This stage was rich of liquid water with LWC up to 0.27 g m$^{-3}$ at a colder temperature (−11 ºC), while there was no appreciable IWC measured in the region (Fig. 4d, e)."

*Lines 183-184. The last sentence in this paragraph doesn't make sense.*

The sentence is deleted in the revised manuscript.

*Lines 188-190. Again, there is no evidence.*

It has been revised:

P7, Line 211: "The clouds in P2 were primarily composed of ice water, with the number concentration of cloud droplets significantly lower compared to P1."

*Line 195. No evidence of "consumed".*

It has been revised:

P7, Line 216: "The lower cloud optical depth and cloud-top temperature in Zhangjiakou area suggested the lower water content and higher cloud-top height of the cloud region."

*Lines 195-196. There is no evidence of "vigorous development of the precipitating cloud".*

It has been deleted in the revised manuscript.

***Lines 209-210. A more accurate statement is that there was an increase in N_round at two levels. Fig 3 suggests one of those might be in a region of low LWC and N_FCDP, and higher concentration of ice particles.***

This suggestion is now added:

P8, Line 230: "In developing cells, $N_{\text{FCDP}}$ tended to decrease with the increase of height, while the diameter of droplets tended to increase (Fig. 6a1), and there was an increase in $N_{\text{Round}}$ at two levels. The broadened droplet spectrum at two levels of developing cells was also observed (Fig. S5)."

***Lines 221-222. What is the evidence for the statement.***

The evidence is now added:

P8, Line 245: "The ice habits were consistent with the feature of cloud region where SIP is thought to be active (Field et al., 2016), considering the temperature of the environment was within the H–M zone, and the region was rich in supercooled large droplets, the H-M process was most likely active (Crosier et al., 2013; Taylor et al., 2016)."

***Lines 222-225. It should be remembered that the pass through the cloud regions are snapshots in time. There is no evidence of "... leading to more small ice through the H-M process...". It is only a suggestion. There is history to consider with vertical and horizontal transport.***

We thank reviewer's suggestion and make revision accordingly on this statement:

P8, Line 251: "This might suggest the great number of large ice at P1.1 improved the riming efficiency and increased the riming surface area, leading to more small ice through H-M process and resulting in the consumption of the droplets. However, the dynamic vertical or horizontal transported of produced ice might induce some uncertainty when evaluating the concentration at the supposed same aircraft position."

**References**

Crosier, J., Choularton, T. W., Westbrook, C. D., Blyth, A. M., Bower, K. N., Connolly, P. J., Dearden, C., Gallagher, M. W., Cui, Z., and Nicol, J. C.: Microphysical properties of cold frontal rainbands†, Quarterly Journal of the Royal Meteorological Society, 140, 1257-1268, 10.1002/qj.2206, 2013.

Field, P. R., Lawson, R. P., Brown, P. R. A., Lloyd, G., Westbrook, C., Moisseev, D., Miltenberger, A., Nenes, A., Blyth, A., Choularton, T., Connolly, P., Buehl, J., Crosier, J., Cui, Z., Dearden, C., DeMott, P., Flossmann, A., Heymsfield, A., Huang, Y., Kalesse, H., Kanji, Z. A., Korolev, A., Kirchgaessner, A., Lasher-Trapp, S., Leisner, T., McFarquhar, G., Phillips, V., Stith, J., and Sullivan, S.: Secondary Ice Production - current state of the science and recommendations for the future, Meteorological Monographs, 10.1175/amsmonographs-d-16-0014.1, 2016.

Heymsfield, A. J., Dye, J. E., and Biter, C. J.: Overestimates of Entrainment From Wetting of Aircraft Temperature Sensors in Cloud, Journal of Applied Meteorology and Climatology, 18, 92-95, 1979.

Holroyd, E. W.: Some Techniques and Uses of 2D-C Habit Classification Software for Snow Particles, Journal of Atmospheric and Oceanic Technology, 4, 498–511, 1987.

Korolev, A. and Isaac, G. A.: Relative humidity in liquid, mixed-phase, and ice clouds., Journal of the atmospheric sciences, 63, 2865-2880, 2006.

Lawson, R. P. and Cooper, W. A.: Performance of some airborne thermometers in clouds, Journal of Atmospheric and Oceanic Technology, 7, 480–494, 10.1175/1520-0426(1990)007,0480:POSATI.2.0.CO;2, 1990.

Lawson, R. P. and Rodi, A. R.: A New Airborne Thermometer for Atmospheric and Cloud Physics Research. Part I: Design and Preliminary Flight Tests, Journal of Atmospheric and Oceanic Technology, 9, 556-574, 1992.

Lawson, R. P., Woods, S., and Morrison, H.: The Microphysics of Ice and Precipitation Development in Tropical Cumulus Clouds, Journal of the Atmospheric Sciences, 72, 2429-2445, 10.1175/jas-d-14-0274.1, 2015.

Lenschow, D. H. and Pennell, W. T.: On the measurement of in-cloud and wet-bulb temperatures from an aircraft, Monthly Weather Review, 102, 447-454, 1974.

Taylor, J. W., Choularton, T. W., Blyth, A. M., Liu, Z., Bower, K. N., Crosier, J., Gallagher, M. W., Williams, P. I., Dorsey, J. R., Flynn, M. J., Bennett, L. J., Huang, Y., French, J., Korolev, A., and Brown, P. R. A.: Observations of cloud microphysics and ice formation during COPE, Atmospheric Chemistry and Physics, 16, 799-826, 10.5194/acp-16-799-2016, 2016.

---

## Author Comment (AC2)

We thank reviewers for their important comments to improve our manuscript. We have now addressed all of the comments raised in the revision. The main revisions we made include:

1. We have avoided the discussions about evolution of clouds, and now focus on the discussion and comparison of cloud microphysics at different stages. The ice mass fraction (IWC/TWC) is now used to indicate the different development stages of clouds, by considering a more mature cloud has a more glaciated fraction.
2. We have carefully revised and responded to the related parts which reviewers consider to be speculative.
3. We have added the radar times and the flight time windows for the four stages, and the time windows of targeting analysis periods.
4. We have carefully examined the manuscript for editorial and grammatical errors, addressing them to enhance the overall readability of our work.

In addition, the relative development stages of clouds are now renamed as P1, P2, P3, and P4 according to the reviewer2's comment. The related descriptions and figures have also been amended in the revised manuscript.

**Reviewer #2**

*This manuscript presents airborne cloud microphysical measurements measured in a mid-latitude extratropical cyclone over China. The authors use the data to explore mechanisms responsible for ice production in different regions of the cloud field and make efforts to link the observed differences to the temporal evolution of the microphysical properties. They show compelling evidence of active secondary ice processes (SIP) in the cloud studied and I particularly liked the section on the production rate of secondary ice. That said, I do have some significant concerns about the analysis that I feel the authors need to address before this manuscript can be considered for publication.*

We thank reviewer for the important comments, we have carefully addressed your comments and have made the following revisions to our manuscript.

**Major comments**

***The authors need to provide evidence that the observations from different regions of the cloud field are showing the temporal microphysical evolution of the cloud microphysics, rather than just presenting measurements that simply document the horizontal variability of cloud properties in the wider cloud field i.e. effectively measuring different clouds. This is key to how the discussion of the observations in the paper is structured, and I am not convinced that the data can be linked together in the way the authors propose. As a result, many of the discussion points made in the paper are speculative. I did wonder if using the ground-based radar measurements to track the temporal evolution of the clouds sampled by the aircraft (before and after the aircraft measurements) might at least enable the airborne data to be put into better context with the "local" cloud development.***

We thank reviewer's comments. The clouds observed were in the same cyclonic system, where the clouds underwent continuous generation, development and dissipation. The microphysical properties under different development stages of clouds are therefore different. However, though in the same synoptic system, it is difficult to explicitly rule out the "evolvement" from the same cloud, because we were measuring different clouds. We therefore now avoid the discussions about evolution of clouds, but using the ice mass fraction (IWC/TWC) to indicate the different development stages of clouds (Fig. 2), by considering a more mature cloud has a more glaciated fraction, for the discussions of cloud microphysics at different stages. The related parts are also amended in the revised manuscript.

Newly added Figure 2:

[Figure]

Figure 2: Ice mass fraction ($F_{Ice}$) as a fuction of total water content at four stages.

Related discussions are added:

P5, Line 156: "Four relative stages during the lifecycle of clouds were identified during experiment, which corresponded to the developing (P1), mature (P2), dissipating (P3) and young cells (P4) in cloud system, according to the different glaciation extents of clouds. The ice mass fraction ($F_{Ice}$: IWC/TWC) was used to indicate the different development stages of clouds (Fig. 2), by considering a more mature cloud has a more glaciated fraction, for the discussions of cloud microphysics at different stages. The cloud system was formed through the combined effects of dynamic forcing induced by the frontal uplift and the moisture transport provided by the prefrontal southerly air mass. Therefore, this study postulated that the continuous clouds within the cloud system had similar dynamic and thermodynamic properties."

*Are these clouds best described as stratocumulus as stated in the title and various other parts of the manuscript? There certainly seems to be convection embedded in the cloud field e.g. updrafts of 10m/s in Fig 3. Would convection embedded in widespread (post-frontal or frontal?) stratiform cloud be a better description? It might be useful to see some satellite imagery of the cloud field.*

We thank reviewer to point this out, and the cloud type is now changed to "stratiform clouds with embedded convection".

The title is changed as:

Microphysical view of development and ice production of mid-latitude stratiform clouds with embedded convection during an extratropical cyclone.

The related parts in manuscript are also revised.

**Additional comments**

*Line 57: What is meant by "on top of the convective core"?*

This is now revised as:

P2, Line 55: "Supercooled large drop may play important roles in the SIP process, which can fracture when freezing and emit ice splinters (Lawson et al., 2015), and this process could extend the SIP to a lower temperature under the influence of strong updraft."

*Line 99: Is the spatial resolution of 1km in the horizontal? If yes, what is the vertical resolution at the typical aircraft location?*

The 1km resolution is in the radial direction, and the vertical resolution of the radar profile along flight track is 30 m (Fig. 4). This is now clarified in the revision:

P4, Line 102: "which can detect targets within a radius of 230 km with a time and radial spatial resolution of 6 minutes and 1 km, respectively."

*Line 106: How good is the circularity threshold of 1.2 on removing out of focus drops i.e. as those show in the imagery in Fig 6? Have the authors performed any visual examination of particles classed as irregular for example?*

The out-of-focus round particles have been corrected following the method by Korolev (2007). The 2D-S images have been visually examined during the data analysis, which are also shown for typical cases (Fig. 7).

*Line 116: Is a different M-D relation used to calculate IWC for the different habits?*

Yes, the different approximated mass formulas for different habits are used to calculate IWC (Holroyd, 1987).

*Line 119: Do the authors use the PCASP data for the calculation of INP? If so, where are these measurements located in relation to the cloud microphysics measurements?*

Yes, the PCASP data measured at the cloud base is used to calculate INP. This is now clarified in the revision:

P5, Line 129: "In this study, $n_{aer,0.5}$ measured by PCASP at the cloud base was used for calculation."

*Fig 1: The caption refers to a blue line, but there is an orange line on the figure.*

The Figure 1 is corrected now, and the relative humidity is indicated by green line in Fig. 1b:

[Figure]

Figure 1: Synoptic overview during experiment. (a) The 500hPa temperature (color), height field (contour), wind field (arrow) at 08:00 (UTC+8h) on September 26[th], 2017; (b) 850hPa divergence field (color), relative humidity (green line, only >80% is shown). The experimental region is indicated by the red dot on each plot.

*The authors refer to both figures in the supplement as e.g. Fig. S1, S2,…etc and stages of the cloud development as S1, S2,….etc. I suggest that the authors differentiate these in any revision.*

The relative development stages of clouds are renamed as P1, P2, P3, P4 now. The related description and figure are also amended in the revised manuscript.

*Line 142-143: It is stated that "aircraft observation area was situated behind the cold front" and "aircraft sampled clouds formed….before the surface cold front". These seem to say the opposite thing. Clarification is needed.*

It is corrected now:

P5, Line 151: "The aircraft sampled the clouds formed in this cyclonic system at this stage, i.e. behind the surface cold front line (Fig. S3) and for the newly formed, developing and matured clouds."

*Line 149: Give more detail on how the different stages are defined.*

The ice mass fraction ($F_{Ice}$: IWC/TWC) is now used to define different development stages of clouds (Fig. 2), by considering a more mature cloud has a more glaciated fraction. The related discussions are added:

P5, Line 158: "The ice mass fraction ($F_{Ice}$: IWC/TWC) was used to indicate the different development stages of clouds (Fig. 2), by considering a more mature cloud has a more

glaciated fraction, for the discussions of cloud microphysics at different stages."

***Fig 2: Can you indicate the times of the radar data on the figure? And what altitude is the reflectivity data from? Is it at the height of the aircraft data in each stage or is it at a fixed altitude?***

The times of the radar plots (Fig. 3 in the revision) and the start/end time for each stage are added in Supplement (Table S1). The radar data used in Fig. 3 is the composite reflectivity data integrated over all altitudes.

Newly added Table S1 (Supplement):

Table S1. Radar times in Fig. 3 and the corresponding flight time windows for the four stages (all in UTC+8h).

| Stages | Radar time | Flight time window | Targeting periods |
|---|---|---|---|
| Developing cells (P1) | 10:06 | 10:03:35-10:23:55 | 1.1: 10:09:13-10:10:31
1.2: 10:11:54-10:12:41 |
| Mature cells (P2) | 10:42 | 10:25:35-11:05:17 | 2.1: 10:42:02-10:42:58
2.2: 10:45:47-10:46:25
2.3: 10:48:38-10:49:33 |
| Dissipating cells (P3) | 11:30 | 11:13:01-11:45:20 | \ |
| Young cells (P4) | 12:00 | 11:46:16-12:15:52 | 4.1: 11:47:16-11:47:35
4.2: 11:51:02-11:51:05
4.3: 11:56:30-11:56:50 |

The related descriptions are added:

P6, Line 164: "The flight tracks mapping on the composite reflectivity of precipitation radar are shown in Fig. 3, colored by the LWC from FCDP and IWC from 2D-S respectively, the radar times and the flight time windows for the four stages are shown in Table S1."

***Line 154: What does "evolved with almost opposite trend" mean?***

It is revised:

P6, Line 166: "In developing cells, substantial LWC was detected up to 0.3 g m$^{-3}$, and the aircraft penetrated a high IWC region in this cloud at 10:09-10:11 BJT, with the highest IWC exceeded 2 g m$^{-3}$ (Fig. 3a1, b1), and $F_{Ice}$ at this stage could span from zero (pure water) to unit (pure ice) (Fig. 2)."

***Line 160: What does "can tell the location of aircraft in cloud" mean?***

This is now revised as:

P6, Line 176: "The cross section of radar reflectivity in Fig. 4a can provide information about the relative positions of aircraft with respect to the cloud top and base, as well as the echo intensity of the cloud."

***Fig 3: What is the uncertainty in the vertical velocity (w) data shown in Fig 3. When looking at the time-series, there seems to be a general negative bias in w. Were any level runs out of cloud performed to see if there was an offset? Also, the uncertainty in these types of measurements is often large when aircraft are not flying straight and level, and Fig 2 shows that there were several large turns and profiles made during the flight. Has this data been quality-checked?***

The AIMMS-20 probe equipped on the aircraft of this experiment is calibrated once every two years according to the procedure by the manufacturer. The vertical wind data during aircraft turns have been removed and are not used for data analysis. There was no offset for vertical wind speed when out of cloud, as the figure below shows.

[Figure]

*Line 168: How sensitive is the fraction of smaller ice to the 180 micron threshold?*

The sensitivity was tested by altering the threshold from 160-200 μm, and the resultant difference of smaller ice fraction was within 10%. This is added in the revision:

P6, Line 186: "The sensitivity was tested by altering the threshold from 160-200 μm, and the resultant difference of smaller ice fraction was within 10%."

*Line 171: States that S2 is the most "vigorously developed clouds", yet the largest updrafts and downdrafts were in S1.*

The stage is now classified as the ice fraction as P2 was more glaciated than P1, but the updraft is not the only criteria. The statement is also revised now:

P7, Line 197: "The cloud-top height in P2 reached 10 km (Fig. 4a), which was the highest cloud-top of clouds observed during experiment. The LWC in P2 was considerably lower compared to P1, while there were more large droplets and ice particles in clouds (Fig. 4d, e), and the distribution of large droplets and ice particles in P2 showed a bimodal distribution. The strength of turbulence in P2 was weaker than P1 (Fig. 4c), but P2 was more glaciated than P1 with $F_{Ice}$ spanning from 0.36 to 1 (Fig. 2)."

*Line 175: The statement of consumption of liquid water in producing ice in the downdraft region is speculative. Could this just be ice precipitation from above?*

The discussions regarding the evolution of clouds (such as consumption of liquid water here) are now removed. The high ice number (> 170 L$^{-1}$) corresponds with the updraft region, thus the ice was likely to be from the layer below. The related discussions in this paragraph are revised:

P6, Line 192: "The ice number peaked at a valley between two peaks of liquid water, but it was difficult to determine the vertical wind at the peak of ice number due to aircraft turns (Fig. S4). However, in the subsequent level flight, high ice number concentration (> 170 L$^{-1}$) was also observed in the strong updraft region. After the high ice number region, LWC up to 0.28 g m$^{-3}$ was observed in the region with weaker updraft."

*Line 177: Statements such as "The droplets at S1 grew to large droplets and were consumed by ice at S2 during the development of cloud" are speculative. Unless it can be demonstrated that the clouds measured at S1 were advected into the region of the measurements at S2 using e.g. trajectories, then these measurements cannot be considered to have been made in the same cloud.*

Similar to above, the "consumed" is now removed. We have now removed all discussions regarding the evolution of clouds. The related discussions are revised:

P7, Line 198: "The LWC in P2 was considerably lower compared to P1, while there

were more large droplets and ice particles in clouds (Fig. 4d, e), and the distribution of large droplets and ice particles in P2 showed a bimodal distribution. The strength of turbulence in P2 was weaker than P1 (Fig. 4c), but P2 was more glaciated than P1 with $F_{Ice}$ spanning from 0.36 to 1 (Fig. 2)."

*Line 183: The measurements with the high drop concentration were also made at warmer temperatures ~ -3C and so it is perhaps not surprising that no ice was measured.*

The descriptions are revised:

P7, Line 205: "This stage was rich of liquid water with LWC up to 0.27 g m$^{-3}$ at a colder temperature (−11 ºC), while there was no appreciable IWC measured in the region (Fig. 4d, e)."

*Paragraph at line 185: Speculation in statements linking different clouds to stages of development.*

The inappropriate statements have been deleted, the authors now focus on the difference in glaciated fraction and characteristics of different clouds, and avoiding linking the microphysical evolution of different stages or different cloud regions. The related discussions are amended:

P7, Line 211: "The clouds in P2 were primarily composed of ice water, with the number concentration of cloud droplets significantly lower compared to P1. P3 was identified as dissipating cells, when the clouds were dominated by ice water, and had a higher $F_{Ice}$ than P2 (Fig. 2)."

*Line 192/Fig S5: The MODIS satellite imagery shows that there was large variability in cloud properties over the region sampled by the aircraft, which again highlights that it is not straightforward to link the observations in terms of stages of cloud development.*

We have now avoided the discussions about evolution of clouds, but using the ice mass fraction (IWC/TWC) to indicate the different development stages of clouds (Fig. 2), by considering a more mature cloud has a more glaciated fraction, for the discussions of cloud microphysics at different stages.

*Figure 6: The overlap between the FCDP and 2DS measurements is poor in the majority of example size distributions. Do the authors know why this is the case?*

The particle size from the 2D-S measurement is determined by the image of casted shadow, previous studies showed this was biased larger for diameter smaller than 30 μm, especially in colder temperatures when some small irregular ice is present (Woods et al., 2018; Gurganus and Lawson, 2018). The FCDP is an optical counter based on

Mie-scattering thus has more accuracy for spheres. Previous studies also observed larger size determined by the 2DS than FCDP(Crosier et al., 2011; Lawson et al., 2006), and the agreement is largely improved for liquid clouds (the following figure is an example). This has been added in the revision:

P8, Line 238: "In addition, the larger size determined by the 2D-S than FCDP was found in Fig. 7, which was due to the lower accuracy for 2D-S to determine the particles in smaller bins (Gurganus and Lawson, 2018; Woods et al., 2018). This may be particularly the case when some small non-spherical ices were present at colder temperatures."

[Figure]

Figure: Particle size distribution of a liquid cloud

*Figure 6: There are many examples of out-of-focus drops (circles with holes in the centre). How were these handled in the processing of 2DS data?*

The out-of-focus round particles have been corrected following the method by Korolev (2007) during data processing.

*Line 217 and the INP spectra in Fig S6. Is this calculated from the Equation on page 4 using the PCASP aerosol concentration measured, and then increased by a factor of 10 to account for uncertainty in the measurements of Demott? And does it therefore represent a likely upper limit on primary INP concentrations?*

Yes, the authors multiply the calculated INP concentration by 10 to represent the upper limit of primary INP concentration. The factor of 10 was pointed out by (Demott et al., 2010) that "This new relationship reduces unexplained variability in ice nuclei concentrations at a given temperature from $\sim 10^3$ to less than a factor of 10, with the remaining variability apparently due to variations in aerosol chemical composition or other factors." And this factor has been used in previous studies to add additional constraining on the determination of secondary ice (Taylor et al., 2016; Sotiropoulou et al., 2020). By considering the potential 10-folds uncertainties, we still found the

concentration of ice exceeded the calculated primary INP concentration, therefore this ensures the SIP identification.

***Line 233: but you do not know where this ice was generated and if it had been transported from other parts of the cloud e.g. that could have been in the H-M zone.***

We agree with the reviewer's comment, while it is unlikely to separate the source of ice particles. The ice particles observed at the same level may involve the particles transported from other parts, and the particles generated by the ice production process. However, the ice production process also involves both the already-formed and ongoing particles, which is a continuous process, and what we have observed or modelled is a net production of ice.

***Line 240: what upper layer?***

This is clarified:

P9, Line 267: "The ice particles observed at this stage most likely originated from the ice nucleating process and the ice falling from above."

***Line 260: Is DCT just a proxy for location with respect to convective cores? And if so, does it just illustrate the microphysical processes in the convection are different to the more widespread stratiform cloud? If so, I might expect a correlation between DCT and updraft strength or turbulence, but it is not obvious that is the case from Fig 3.***

The DCT represents the location of observed region relative to the cloud-top, but may not be only limited to convective clouds. The clouds in this study have included both widespread stratiform and imbedded convective clouds, and the metric of DCT should all apply. The DCT essentially implies the amount of ice hydrometeors may fall from above, but may not be directly associated with updraft strength or turbulence. This is now added:

P10, Line 301: "It should be noted that the observed clouds have included both widespread stratiform and imbedded convective clouds, and the metric of DCT should all apply. The DCT essentially implies the amount of ice hydrometeors may fall from above, but may not be directly associated with updraft strength or turbulence."

***Line 269: Again, speculation.***

This is now revised:

P10, Line 299: "This suggested the development of cloud-top increased $N_{Ice}$, and considering the larger particles tended to fall to cloud base and form precipitation, the reduced $N_{Ice}$ close to cloud base may be due to the coalesce of ice which reduced the number but enlarged the size of ice."

*Line 275: But the aircraft is measuring different clouds and so there could be many reasons why the ice concentration is different from penetrations made at the same height.*

We agree with reviewer that many factors could lead to different ice concentration but the DCT is a factor more apparently influencing the ice production based on the observation here.

*Line 312: droplet > 25 microns?*

This is clarified:

P12, Line 359: "Considering that the observation here was actually after the SIP process was initialized, when the smaller cloud droplets had been considerably consumed and most graupels were rimed, the number of large droplets $(d > 50\ \mu m)$ was the limited factor for SIP, and therefore used to calculate the modelled SIP rate."

*Line 322: it is assumed that all ice is graupel, but in Fig 6 the habit classification shows that plates are the dominant habit.*

Upon visual observation, most ice particles exhibit obvious riming characteristics, particularly larger ones, including plate ice, as shown in Fig. 6 (Fig. 7 in the revision). Therefore, ice particles larger than 250 μm are primarily considered as graupels.

*Line 352: I think this is speculative.*

This is revised with more evidence referenced.

P10, Line 314: "The explanation was also similar to the results reported by Li et al. (2021), which showed the columnar ice crystals were produced in the lower layer seeded by ice particles falling from the upper layer."

*Line 364: The last sentence is rather generic. Can the authors provide some more information on how these measurements could be used to "improve the understanding of key processes" and "help find the region of supercooled water of clouds for the weather modification work".*

This is now added:

P13, Line 403: "The results about the microphysical properties of stratiform clouds with convective cells under different stages suggest the falling hydrometeors associated with cloud-top height importantly controlled the cloud glaciation and precipitation process, and this may also help find the region of supercooled water of clouds for the weather modification work."

***Finally, there are many instances where the English text could be improved on, and this is something that the reviewers should also try to address in any revision.***

We have carefully examined the manuscript for editorial and grammatical errors, addressing them to enhance the overall readability of our work.

**References**

Crosier, J., Bower, K. N., Choularton, T. W., Westbrook, C. D., Connolly, P. J., Cui, Z. Q., Crawford, I. P., Capes, G. L., Coe, H., Dorsey, J. R., Williams, P. I., Illingworth, A. J., Gallagher, M. W., and Blyth, A. M.: Observations of ice multiplication in a weakly convective cell embedded in supercooled mid-level stratus, Atmospheric Chemistry and Physics, 11, 257-273, 10.5194/acp-11-257-2011, 2011.

DeMott, P. J., Prenni, A. J., Liu, X., Kreidenweis, S. M., Petters, M. D., Twohy, C. H., Richardson, M. S., Eidhammer, T., and Rogers, D. C.: Predicting global atmospheric ice nuclei distributions and their impacts on climate, Proc Natl Acad Sci U S A, 107, 11217-11222, 10.1073/pnas.0910818107, 2010.

Gurganus, C. and Lawson, P.: Laboratory and Flight Tests of 2D Imaging Probes: Toward a Better Understanding of Instrument Performance and the Impact on Archived Data, Journal of Atmospheric and Oceanic Technology, 35, 1533-1553, 10.1175/jtech-d-17-0202.1, 2018.

Holroyd, E. W.: Some Techniques and Uses of 2D-C Habit Classification Software for Snow Particles, Journal of Atmospheric and Oceanic Technology, 4, 498–511, 1987.

Korolev, A.: Reconstruction of the Sizes of Spherical Particles from Their Shadow Images. Part I: Theoretical Considerations, Journal of Atmospheric and Oceanic Technology, 24, 376-389, 10.1175/jtech1980.1, 2007.

Lawson, R. P., Woods, S., and Morrison, H.: The Microphysics of Ice and Precipitation Development in Tropical Cumulus Clouds, Journal of the Atmospheric Sciences, 72, 2429-2445, 10.1175/jas-d-14-0274.1, 2015.

Lawson, R. P., O'Connor, D., Zmarzly, P., Weaver, K., Baker, B., Mo, Q., and Jonsson, H.: The 2D-S (Stereo) Probe: Design and Preliminary Tests of a New Airborne, High-Speed, High-Resolution Particle Imaging Probe, Atmospheric and Oceanic Technology, 23, 1462–1477, 2006.

Li, H., Möhler, O., Petäjä, T., and Moisseev, D.: Two-year statistics of columnar-ice production in stratiform clouds over Hyytiälä, Finland: environmental conditions and the relevance to secondary ice production, Atmospheric Chemistry and Physics, 21, 14671-14686, 10.5194/acp-21-14671-2021, 2021.

Sotiropoulou, G., Sullivan, S., Savre, J., Lloyd, G., Lachlan-Cope, T., Ekman, A. M. L., and Nenes, A.: The impact of secondary ice production on Arctic stratocumulus, Atmospheric Chemistry and Physics, 20, 1301-1316, 10.5194/acp-20-1301-2020, 2020.

Taylor, J. W., Choularton, T. W., Blyth, A. M., Liu, Z., Bower, K. N., Crosier, J., Gallagher, M. W., Williams, P. I., Dorsey, J. R., Flynn, M. J., Bennett, L. J., Huang, Y., French, J., Korolev, A., and Brown, P. R. A.: Observations of cloud microphysics and ice formation during COPE, Atmospheric Chemistry and Physics, 16, 799-826, 10.5194/acp-16-799-2016, 2016.

Woods, S., Lawson, R. P., Jensen, E., Bui, T. P., Thornberry, T., Rollins, A., Pfister, L., and Avery, M.: Microphysical Properties of Tropical Tropopause Layer Cirrus, Journal of Geophysical Research: Atmospheres, 123, 6053-6069, 10.1029/2017jd028068, 2018.

---

## Referee Report (RR1)

The authors have provided good responses to most of the reviewers' comments and have made appropriate changes to most of the manuscript. There remains some speculation that should be stated as such. Details are given in the comments below.

Some of the responses to reviewers' comments require clarification.

There are several instances where the English text could be improved upon. Many of the mistakes come from incorrect word use.

**Comments on responses to the Reviewer 2's comments**

**Line 99** *Is the spatial resolution of 1km in the horizontal? If yes, what is the vertical resolution at the typical aircraft location?*

What does it mean that the vertical resolution of the radar profile along flight track is 30 m (Fig. 4)? Has interpolation been done? The beam width will be significant at a range of 50-200 km.

**Line 217 and the INP spectra in Fig S6** *Is this calculated from the Equation on page 4 using the PCASP aerosol concentration measured, and then increased by a factor of 10 to account for uncertainty in the measurements of Demott? And does it therefore represent a likely upper limit on primary INP concentrations?*

Make it clear in Figure S7 caption that the number concentration has been multiplied by a factor of 10 ..."to represent a likely upper limit on primary INP concentrations"?

**Line 233** *but you do not know where this ice was generated and if it had been transported from other parts of the cloud e.g. that could have been in the H-M zone.*

Has there been any change to the text in response to this comment? It isn't clear what is meant by observing "a net production of ice."

**Comments on the revised paper**

**Line 253** *However, the dynamic vertical or horizontal transported of produced ice might induce some uncertainty when evaluating the concentration at the supposed same aircraft position.*

This does not describe the problem. It would be useful to describe a possible scenario of ice being transported from e.g near cloud top by the circulations in convective thermals.

**Line 260** *Although the average temperature of P2.1 was as low as -11.7C the abundant large ice particles triggered the active SIP process at P2.1 with high NIce about 300 L-1, indicating that the SIP process was not restricted by temperature.*

This is speculation. There is no direct evidence that the large ice particles triggered the active SIP process. The fact that there is a high concentration of ice particles may be due to vertical and/or horizontal transport previous to the time the pass was made.

**Line 262** Similarly, the history of development leading to the cloud regions in P2.2 and P2.3 cannot be determined, can it? I suggest that sentence be modified or deleted.

**Line 265** *Aircraft penetrated the cloud-top at P4.3, where several primary ice particles could be observed (Fig. 7c).*

It can only be speculation that these are primary ice particles.

**Line 269** *The size spectrum and 2D-S images in Fig. 7c showed that large ice presented at P4.1, and ice grew through riming and Bergeron processes, while the ice at P4.2 was mainly smaller ice, which was still in the process of growth.*

Again, it is speculation that the smaller ice was still in the process of growth. Also, the wording of the first part of the sentence should be corrected.

**Line 272** *The large ice falling from the upper layer played a very important role in ice production process, the primary ice crystals formed through the nucleation process and grew up in the upper layer or during the fall, then fell to the lower layer to trigger the ice production process.*

This should be stated as the likely or possible process.

**Line 276** *There was a positive correlation between NIce and NRound, where more large droplets generally corresponded to a higher NIce.*

Considering Reviewer 2's comment: There are many examples of out-of-focus drops (circles with holes in the centre). How were these handled in the processing of 2DS data. And the authors' response: The out-of-focus round particles have been corrected following the method by Korolev (2007) during data processing.

How are the out-of-focus non-circular images handled? Is it possible that many of the images with holes are not drops, but larger out-of-focus ice particles?

**Line 303** "...but may not be directly associated with *the current* updraft strength or turbulence."

**Line 308** *This suggested the DCT tended to be a more important factor than temperature.*

It is possible of course that the cloud top had ascended to a greater altitude, but then collapsed by the time of the observation.

It seems wrong to say that DCT is more important than temperature in the HM temperature zone. Most likely they are of similar importance.

**Figure 10** Is it not likely that graupel particles will pass through the HM zone in Mature cells and so SIP will occur in the HM zone? It should be emphasised that the discussion on p10 that concerns Fig 10 is a possible scenario (or likely?). There is no direct evidence.

**Line 391** *The seeder-feeder process was found...*

It is only a possible explanation.

---

## Author Response (AR2)

We once again thank reviewers for their comprehensive review and comments on our manuscript. We have now addressed the remaining issues reviewers raised.

**Reviewer #1**

*The authors have provided good responses to most of the reviewers' comments and have made appropriate changes to most of the manuscript. There remains some speculation that should be stated as such. Details are given in the comments below. Some of the responses to reviewers' comments require clarification.*
*There are several instances where the English text could be improved upon. Many of the mistakes come from incorrect word use.*

We appreciate the important comments from reviewer. We have addressed all of the comments raised in the revision, clarified related responses, and carefully reviewed the manuscript for editing and grammatical errors.

**Comments on responses to the Reviewer 2's comments**

*Line 99 Is the spatial resolution of 1km in the horizontal? If yes, what is the vertical resolution at the typical aircraft location?*
*What does it mean that the vertical resolution of the radar profile along flight track is 30 m (Fig. 4)? Has interpolation been done? The beam width will be significant at a range of 50-200 km.*

Yes, the radial spatial resolution of radar is 1 km, and the radar reflectivity in radar profile is calculated by linear interpolation in nearest neighbor combined with a vertical direction (NVI) method, and the vertical resolution of the radar profile after interpolation is 30 m. Related discussions are revised:

P6, Line 184: "A vertical line was first determined according to the latitude and longitude of the aircraft, then the azimuth angles, elevation angles, and range bins of equidistant points with a resolution of 30 m in the vertical line were obtained, and the radar reflectivity of each equidistant point was calculated by linear interpolation in nearest neighbor combined with a vertical direction (NVI) method. Hereby the radar profile with vertical resolution of 30 m along flight track was obtained."

*Line 217 and the INP spectra in Fig S6 Is this calculated from the Equation on page 4 using the PCASP aerosol concentration measured, and then increased by a factor of 10 to account for uncertainty in the measurements of Demott? And does it therefore represent a likely upper limit on primary INP concentrations?*
*Make it clear in Figure S7 caption that the number concentration has been multiplied by a factor of 10 ..."to represent a likely upper limit on primary INP concentrations"?*

We thank reviewer's comment, and the caption of Figure S7 (Figure S5 in the revision) is revised now:

[Figure]

Figure S5: The number concentration of ice nucleating particles ($N_{INP}$) as a function of cloud temperature, and the $N_{INP}$ has been multiplied by a factor of 10 to represent a likely upper limit on primary INP concentrations.

*Line 233 but you do not know where this ice was generated and if it had been transported from other parts of the cloud e.g. that could have been in the H-M zone. Has there been any change to the text in response to this comment? It isn't clear what is meant by observing "a net production of ice."*

The related discussions are added now:

P9, Line 259: "It should be noted that the observed $N_{Ice}$ may involve hydrometeors transported from other parts of clouds, along with the locally produced ice. The ice production can therefore be considered as a continuous process and the observed $N_{Ice}$ is a net production of ice after considering all the input (local production and transport in) and output (fall out and transport out) factors at the observation level."

**Comments on the revised paper**

*Line 253 However, the dynamic vertical or horizontal transported of produced ice might induce some uncertainty when evaluating the concentration at the supposed same aircraft position.*
*This does not describe the problem. It would be useful to describe a possible scenario of ice being transported from e.g near cloud top by the circulations in convective thermals.*

The discussion is revised according to reviewer's suggestion:

P8, Line 246: "However, the dynamic vertical or horizontal transported of ice, e.g. in

convective thermals, the ice near cloud-top can be circulated downwards surrounding the convection core, while being transported upward in the convection core (Korolev et al., 2020). This might induce some uncertainty when evaluating the concentration at the aircraft observed position."

*Line 260 Although the average temperature of P2.1 was as low as -11.7C the abundant large ice particles triggered the active SIP process at P2.1 with high NIce about 300 L-1, indicating that the SIP process was not restricted by temperature.*
*This is speculation. There is no direct evidence that the large ice particles triggered the active SIP process. The fact that there is a high concentration of ice particles may be due to vertical and/or horizontal transport previous to the time the pass was made.*

It is now revised as:

P8, Line 255: "Although the average temperature of P2.1 was as low as -11.7 °C, the abundant large ice particles seemed to trigger the active SIP process at P2.1 with high $N_{Ice}$ about 300 L$^{-1}$, indicating that the SIP process might not be restricted by temperature, though the possible transport of ice from other cloud regions is not able to be completely excluded."

*Line 262 Similarly, the history of development leading to the cloud regions in P2.2 and P2.3 cannot be determined, can it? I suggest that sentence be modified or deleted.*

It is now modified as:

P9, Line 258: "The period 2.2 that lacked enough large ice was likely still in the glaciation process, and P2.3 might be difficult to trigger a more active SIP process due to the smaller number of large ice and limited liquid water."

*Line 265 Aircraft penetrated the cloud-top at P4.3, where several primary ice particles could be observed (Fig. 7c)*
*It can only be speculation that these are primary ice particles.*

It has been revised:

P9, Line 268: "Aircraft penetrated the cloud-top at P4.3 and observed several ice particles (Fig. 7c), which were likely primary ice particles."

*Line 269 The size spectrum and 2D-S images in Fig. 7c showed that large ice presented at P4.1, and ice grew through riming and Bergeron processes, while the ice at P4.2 was mainly smaller ice, which was still in the process of growth.*
*Again, it is speculation that the smaller ice was still in the process of growth. Also, the wording of the first part of the sentence should be corrected*

The discussions are revised:

P9, Line 269: "The size spectrum and 2D-S images in Fig. 7c showed that large ice particles presented at P4.1, ==and the images suggested these were likely formed== through riming and Bergeron processes, while the ice at P4.2 was mainly smaller ice, ==possibly== still in the process of growth."

*Line 272 The large ice falling from the upper layer played a very important role in ice production process, the primary ice crystals formed through the nucleation process and grew up in the upper layer or during the fall, then fell to the lower layer to trigger the ice production process.*
*This should be stated as the likely or possible process.*

The description is revised:

P9, Line 272: "The large ice falling from the upper level ==likely== played a very important role in the ice production process, ==where== the primary ice crystals ==might== form through the nucleation process and grow up in the upper level or during the fall, then fall to the lower level to trigger the ice production process."

*Line 276 There was a positive correlation between NIce and NRound, where more large droplets generally corresponded to a higher NIce.*
*Considering Reviewer 2's comment: There are many examples of out-of-focus drops (circles with holes in the centre). How were these handled in the processing of 2DS data. And the authors' response: The out-of-focus round particles have been corrected following the method by Korolev (2007) during data processing. How are the out-of-focus non-circular images handled? Is it possible that many of the images with holes are not drops, but larger out-of-focus ice particles?*

The out-of-focus images are because of light diffraction by sphere, which is less likely to occur for non-spherical particles, i.e., ice particles (Mcfarquhar et al., 2017; Vaillant De Guélis et al., 2019). We therefore consider this group of images are mainly from droplets. A number of previous studies also consider these mainly resulted from droplets (Crosier et al., 2011; Lawson et al., 2015; Woods et al., 2018).

*Line 303 "...but may not be directly associated with \*the current\* updraft strength or turbulence."*

It is revised:

P10, Line 302: "The DCT essentially implies the amount of ice hydrometeors may fall from above, but may not be directly associated with ==the current== updraft strength or turbulence."

*Line 308 This suggested the DCT tended to be a more important factor than temperature.*
*It is possible of course that the cloud top had ascended to a greater altitude, but then*

*collapsed by the time of the observation.*
*It seems wrong to say that DCT is more important than temperature in the HM*
*temperature zone. Most likely they are of similar importance.*

Thank reviewer to point this out, and the statement is revised:

P10, Line 308: "This suggested that ==the DCT played an important role in SIP process,==
==and in the region with lower temperature than H-M temperature zone,== the DCT tended
to be a more important factor than temperature in determining the intensity of SIP."

*Figure 10 Is it not likely that graupel particles will pass through the HM zone in*
*Mature cells and so SIP will occur in the HM zone? It should be emphasised that the*
*discussion on p10 that concerns Fig 10 is a possible scenario (or likely?). There is no*
*direct evidence*

The related discussions are amended according to the reviewer's comments:

P10, Line 320: "Then the ==likely== schematic plot of ice production at different stages of
clouds was given (Fig. 10)."

P11, Line 327: "==However, it should be noted that the larger ice particles may also falling==
==to H-M zone in mature cells and trigger the SIP process.=="

*Line 391 The seeder-feeder process was found...*
*It is only a possible explanation.*

It is revised:

P13, Line 394: "The ==possible== seeder-feeder process was found to extend the SIP process
beyond the slightly supercooled temperature region for the typically considered H-M
process."

**Reviewer #2**

**Main comments**

*1. The removal of much of the wording and discussion on the evolution of cloud properties from different measurement periods is a good improvement from the original submission. The authors now use the ratio of IWC/TWC to document differences in the microphysics in clouds with different degrees of glaciation. I think that some additional description in section 3.1 is needed though. i) The authors refer to figure 2 to illustrate how the different periods were identified for subsequent analysis. However, periods P2 and P3 look very similar and so using this metric alone does not seem to be sufficient. Consider adding some text that provides more information on how the periods were selected. ii) Line 161: "this study postulated that the continuous clouds within the cloud system had similar dynamic and thermodynamic properties". This seems to be the key sentence that the authors use to justify the discussion about the "development" of clouds within the larger system, based on the glaciation metric. Please add some additional justification for this assumption. Are there previous studies that follow this approach in similar cloud systems for example?*

We thank the positive comments from reviewer on our current manuscript. We have now addressed the remaining comments reviewer raised.

A paragraph is now added after the discussions for P2 and P3:

P6, Line 162: "Although the glaciation extents between P2 and P3 were similar, P3 showed a narrower cloud band (Fig. 3) and a lower cloud-top (Fig. 4) for dissipating cells compared to the mature clouds in P2."

P6, Line 165: "Therefore, this study postulated that the continuous clouds within the cloud system had similar dynamic and thermodynamic properties. Previous studies also pointed out the exchangeability between temporospatial domains of cloud properties in the same cloud system, where properties and evolution of individual clouds were similar (Lensky and Rosenfeld, 2006; Yuan et al., 2010; Coopman et al., 2020)."

*2. The break-down of figure 4 into multiple panels is very nice as it allows the reader to examine the data from the different periods more easily. I would suggest using the new figure (Fig S4) instead of Fig 4 in the main paper. You could always put the current Fig 4 in the supplement if needed.*

We thank the reviewer's comment, and Fig. S4 has been used instead of Fig. 4 in the revised manuscript, and Fig. 4 is deleted now.

*3. Consider removing the MODIS satellite analysis (paragraph beginning line 214, Fig S6), as the satellite data is just a single snapshot in time, and so it is not straightforward to link it to the different time-periods from the aircraft data. Also, it*

*is worth noting that the satellite cloud effective radius data is at cloud-top and so not necessarily comparable to microphysics measurements made lower down in the clouds. As per my original review though, I do think it would be more useful to see a satellite image (visible or IR) of the cloud field presented early on in the text, to give the reader a better sense of the cloud system studied.*

Fig. S6 and related analysis have been removed in the revision considering this suggestion.

*4. In the description of the microphysical measurements the authors use the terminology of cloud "cells" and cloud "layers" e.g. "developing cells" and "ice particles fell from the upper layer to the lower layer". Can you clarify how you differentiate a cloud "cell" from a cloud "layer" in the measurements? It cannot solely be on the measured updraft strength, as period P4 is characterized as "young cells" but there are no significant updrafts shown in Fig S4d. Is it based on the radar echoes? Or something else?*

The "cells" is used to denote different clouds discussed in manuscript, while the "layer" is used to indicate different height levels within clouds, not cloud layers. The "upper layer" or "lower layer" described in analysis does not refer to a specific height, but rather broadly denotes higher altitude levels or lower altitude levels within the cloud. In the revised manuscript, the "layer" has been changed to "level" for clarification.

**Additional comments**

*1. Line 70: What do you mean by a "typical mid-latitude cloud"? What features make it "typical"?*

The "typical" is now removed for clarification.

*2. Line 121: Please define what TWC is when you introduce it. I assume it is the sum of the liquid and ice water contents. But it wasn't clear which cloud probes were used to calculate the TWC (FCDP, 2DS, HVPS). And if using a combination of probes, how was this done e.g. considering probe overlap in particle sizing.*

The definition of TWC is added:

P4, Line 123: "The total water content (TWC) was obtained by adding the IWC calculated from the 2D-S (diameter 10-1280 μm) and LWC measured by the FCDP (diameter 2-50 μm)."

*3. Line 130: Please clarify if the PCASP measurements were made below cloud base? From the current text it isn't clear that this is the case. It is important to mention as PCASP measurements in cloud often exhibit artifacts from cloud particles shattering on the inlet.*

These discussions are now added:

P5, Line 130: "In this study, the PCASP measurement was conducted below cloud base, and the in-cloud PCASP data was excluded for analysis due to cloud particle shattering on the inlet. Therefore, $n_{aer,0.5}$ measured by PCASP below cloud base was used for calculation."

**4. Line 186: I think the last two sentences in this paragraph would read better if the order was switched i.e. the sentence beginning "The fraction of smaller…." was before the sentence beginning "The sensitivity was tested…."**

Thank the reviewer' comments, and it is revised now:

P7, Line 192: "The fraction of smaller ice with $d$ < 180 μm ($F_{smaller\ ice}$) was defined to imply the freshly formed smaller ice which had not experienced sufficient growth (Fig. 4b). The sensitivity was tested by altering the threshold from 160-200 μm, and the resultant difference of smaller ice fraction was within 10%."

**5. Line 200: Please clarify what you mean by "turbulence". It is not obvious that measures like the vertical velocity variance are greater in P1 for example. Or do you just mean the peak updraft strength?**

In the revised manuscript, the turbulence was replaced by updraft for clarification:

P7, Line 195: "P1 featured strong updraft with vertical wind speed up to 8.9 m/s, and the strong updraft region was dominated by ice particles and precipitation particles (Fig. 4c-e)."

P7, Line 204: "The updraft strength in P2 was weaker than P1 (Fig. 4c), but P2 was more glaciated than P1 with $F_{Ice}$ spanning from 0.36 to 1 (Fig. 2)."

**6. Line 206: Fig S4d shows that there were measured ice concentrations of 80-120 L-1 in P4. Yet the text states that "there was no appreciable IWC measured in this region".**

Thank the reviewer to point this out, the calculated IWC in this colder temperature region is also significantly lower compared to other stages (Fig. 5), the related discussion is revised:

P7, Line 210: "This stage was rich of liquid water with LWC up to 0.27 g m$^{-3}$ at a colder temperature (−11 ⁰C), while the IWC measured in the region was significantly lower compared to other stages (Figs. 4d, e and 5)."

**7. Line 254: What is meant by "at the supposed same aircraft position"?**

This is more clearly stated:

P8, Line 246: "However, the dynamic vertical or horizontal transported of ice, e.g. in convective thermals, the ice near cloud-top can be circulated downwards surrounding the convection core, while being transported upward in the convection core (Korolev et al., 2020). This might induce some uncertainty when evaluating the concentration at the aircraft observed position."

**8. There are still many instances where the English text could be improved on.**

We have carefully reviewed the manuscript for editing and grammar errors.

**Reference**

Coopman, Q., Hoose, C., and Stengel, M.: Analysis of the Thermodynamic Phase Transition of Tracked Convective Clouds Based on Geostationary Satellite Observations, Journal of Geophysical Research: Atmospheres, 125, 10.1029/2019jd032146, 2020.

Crosier, J., Bower, K. N., Choularton, T. W., Westbrook, C. D., Connolly, P. J., Cui, Z. Q., Crawford, I. P., Capes, G. L., Coe, H., Dorsey, J. R., Williams, P. I., Illingworth, A. J., Gallagher, M. W., and Blyth, A. M.: Observations of ice multiplication in a weakly convective cell embedded in supercooled mid-level stratus, Atmospheric Chemistry and Physics, 11, 257-273, 10.5194/acp-11-257-2011, 2011.

Korolev, A., Heckman, I., Wolde, M., Ackerman, A. S., Fridlind, A. M., Ladino, L. A., Lawson, R. P., Milbrandt, J., and Williams, E.: A new look at the environmental conditions favorable to secondary ice production, Atmospheric Chemistry and Physics, 20, 1391-1429, 10.5194/acp-20-1391-2020, 2020.

Lawson, R. P., Woods, S., and Morrison, H.: The Microphysics of Ice and Precipitation Development in Tropical Cumulus Clouds, Journal of the Atmospheric Sciences, 72, 2429-2445, 10.1175/jas-d-14-0274.1, 2015.

Lensky, I. and Rosenfeld, D.: The time-space exchangeability of satellite retrieved relations between cloud top temperature and particle effective radius, Atmospheric Chemistry and Physics, 6, 2887-2894, 2006.

McFarquhar, G. M., Baumgardner, D., Bansemer, A., Abel, S. J., Crosier, J., French, J., Rosenberg, P., Korolev, A., Schwarzoenboeck, A., and Leroy, D.: Processing of ice cloud in situ data collected by bulk water, scattering, and imaging probes: Fundamentals, uncertainties, and efforts toward consistency, Meteorological Monographs, 58, 11.11-11.33, 2017.

Vaillant de Guélis, T., Schwarzenböck, A., Shcherbakov, V., Gourbeyre, C., Laurent, B., Dupuy, R., Coutris, P., and Duroure, C.: Study of the diffraction pattern of cloud particles and the respective responses of optical array probes, Atmospheric Measurement Techniques, 12, 2513-2529, 10.5194/amt-12-2513-2019, 2019.

Woods, S., Lawson, R. P., Jensen, E., Bui, T. P., Thornberry, T., Rollins, A., Pfister, L., and Avery, M.: Microphysical Properties of Tropical Tropopause Layer Cirrus, Journal of Geophysical Research: Atmospheres, 123, 6053-6069, 10.1029/2017jd028068, 2018.

Yuan, T., Martins, J. V., Li, Z., and Remer, L. A.: Estimating glaciation temperature of deep convective clouds with remote sensing data, Geophysical Research Letters, 37, 10.1029/2010gl042753, 2010.

---

## Author Response (AR3)

We sincerely appreciate Editor's thorough review on the manuscript. The comments are important to improve the quality of our manuscript. We have now addressed the remaining issues. In the revised version, speculative discussions have been reorganized in the conclusion section. Furthermore, the entire manuscript has undergone professional editing to enhance grammar, sentence structure, and overall readability.

**Comments**

*1. I have some difficulty with the fact that your results (which can be supported by direct evidence) are at times mixed with speculations that are made without firm evidence or complete information. An example of this is the speculation on the seeder-feeder process, as noted by Reviewer #1. The text on lines 310-317 is only speculative, but the way it is presented together with the results has given the reader the impression that this is a finding from your analysis. Another example is the schematic diagram shown in Figure 10. Several processes presented in the diagram are also speculative, as pointed out by Reviewer #1, but there have been no distinctions between the actual findings and speculations provided in the manuscript. I believe a more careful categorization/organization of the results and discussion is required. This is also in line with the main concern from Reviewer #1.*

We have now merged the speculative discussions into the conclusion and reorganized the original conclusion. The current discussion and conclusion sections now include a summary of the key results, discussions about comparison to previous studies, and about the limitations. The specific revisions include:

(1) The discussions on the seeder-feeder process explaining the positive correlation between DCT and $N_{Ice}$ have been moved to the "discussion and conclusions" section:

P12, Line 384: "The results revealed generally enhanced SIP when greater distance to cloud-top, which could be explained by the seeder-feeder mechanism occurring in stratiform cloud precipitation (Hobbs and Locatelli, 1978; Hobbs et al., 1980; Matejka et al., 1980): when the cloud-top is higher, more primary ice particles form at colder temperatures and fall. The ice particles can capture smaller liquid water droplets when falling, during which they can grow and the fall speed can be accelerated. This process can considerably enhance the interaction between ice and water droplets or among ice particles, which is necessary for the occurrence of ice fracturing, thereby leading to the avalanche SIP. The age of ice could be estimated on the basis of the fraction of smaller ice ($F_{smaller\ ice}$) here, with the assumption that recently formed ice particles are smaller in size. This implied the pronounced production of smaller ice particles by SIP processes, with $F_{smaller\ ice}$ reaching 70% during the developing period, whereas a lower $F_{smaller\ ice}$ (0.2–0.6) indicated the growth of ice and smaller ice was consumed during the dissipating stage (Fig. 4). This explanation is also similar to the results reported by Li et al. (2021), who reported that columnar ice crystals were produced at the lower level and were seeded by ice particles falling from the upper level."

(2) The diagram of ice production (Fig. 11 in the revised version) and related explanations have been placed after the discussion of the seeder-feeder process, followed by a summary of the key factors controlling the SIP rate.

P13, Line 395: "The likely schematic plot of ice production at different stages of clouds is given in Fig. 11. A higher cloud-top leads to the formation of more primary ice through the nucleation process, and the ice can grow in the upper level and during the fall. The SIP process is triggered when ice particles in the upper level fall to the lower level with supercooled water, initiating the interactions between ice and droplets. In regions with larger DCTs, ice particles in the upper level have sufficient time and distance to grow larger during the fall, and the fall speed can also be accelerated, resulting in more and larger ice particles falling to the lower level. Consequently, the intensity of the SIP process becomes stronger in this region because the falling large ice particles enhance the interactions between ice and droplets, as well as among ice particles. However, larger ice particles may also fall into the H–M zone in mature cells and trigger the SIP process. Moreover, this possible seeder-feeder process was found to extend the SIP process beyond the slightly supercooled temperature region for the typically considered H–M process. The intensity of SIP was to the first order determined by the numbers of graupel and droplets, because the collision and coalescence processes among these hydrometeors necessitated the fracturing of ice. The modelled and measurement-based calculations showed that appropriately treating the size distribution hereby the determination of collection efficiency will improve the modelling of the SIP rate."

(3) The last paragraph of the "discussion and conclusions" section provides a summary of the findings, along with relevant caveats and limitations.

P13, Line 408: "Our results indicate that once the cloud-top reaches a sufficient height, the ice initialized from nucleation may boost the avalanche glaciation process when falling ice reaches lower levels in clouds. It should be noted that whether the falling hydrometeors were the ones generated by the ice production process or were about to participate in the ice production process at the same level, may never be separated due to the short time scale of the collision process. However, this is a continuous process that may involve both already-formed and ongoing-happening particles, and the observed or modelled results are an overall net production of ice. The ice particles falling from aloft increase the number of graupel particles and the chance of collision between graupel and droplets and then trigger the SIP process; therefore, the seeder-feeder and SIP processes may occur simultaneously after the SIP process has initialized. The results concerning the microphysical properties of stratiform clouds with convective cells under different stages suggest that the falling hydrometeors associated with the cloud-top height importantly control the cloud glaciation and precipitation processes, and this information may also help find the region of supercooled water in clouds for weather modification work."

*2. In view of my comment above, you could consider merging the 'discussion' material (which may contain speculative content where appropriate) with the current 'conclusion' section. ACP has provided specific guidelines for the concluding section, which can be found below.*

*https://www.atmospheric-chemistry-and-physics.net/policies/guidelines_for_authors.html#:~:text=ACP%20expects%20that%20the%20concluding,include%20the%20main%20quantitative%20results.*

*In general, ACP expects that this section will normally include a summary, synthesis/interpretation, comparison and context, caveats and limitations. Please consider incorporating all these components in your revised concluding section.*

We appreciate Editor's comments and have revised the conclusion section in accordance with suggestions. The revised conclusion section more compiles with the guidelines of ACP.

*3. As noted by both reviewers, there remain many instances in the current manuscript where the English text could be improved on. You stated in your response to Reviewer #2 that "we have carefully reviewed the manuscript for editing and grammar errors". However, I cannot identify any relevant revisions in your tracked changes. Indeed, in my reading of your current manuscript I have identified several grammatical errors including incorrect comma splices, fragment sentences, combining clauses incorrectly, and incorrect (e.g. non-scientific) word use, etc.. I strongly recommend that you seek professional assistance for copyediting, or thoroughly revise the manuscript to improve grammar, sentence structure, and overall fluency.*

Thank Editor for pointing this out. The grammar, sentence structure, and overall readability of the manuscript have now been improved by professional editing service, and the certificate after the service is attached below.

[Figure]

References

Hobbs, P. V. and Locatelli, J. D.: Rainbands, Precipitation Cores and Generating Cells in a Cyclonic Storm, Journal of Atmospheric Sciences, 35, 230-241, 1978.

Hobbs, P. V., Matejka, T. J., Herzegh, P. H., Locatelli, J. D., and Houze , R. A.: The Mesoscale and Microscale Structure and Organization of Clouds and Precipitation in Midlatitude Cyclones. I: A Case Study of a Cold Front, Journal of Atmospheric Sciences, 37, 568-596, 1980.

Matejka, T. J., Houze, R. A., and Hobbs, P. V.: Microphysics and dynamics of clouds associated with mesoscale rainbands in extratropical cyclones, Quarterly Journal of the Royal Meteorological Society, 106, 29–56, 10.1002/qj.49710644704 1980.